# Numerical Analysis of Friction Reduction and ATSLB Capacity of Lubricated MTS with Textured Micro-Elements

**Xigui Wang** [1,*], **Hui Huang** [1,*], **Jingyu Song** [1], **Yongmei Wang** [1,2] **and Jiafu Ruan** [1,3]

[1] School of Mechatronics and Automation, Huaqiao University, No. 668 Jimei Avenue, Jimei District, Xiamen 361021, China
[2] School of Motorcar Engineering, Heilongjiang Institute of Technology, No. 999, Hongqidajie Road, Daowai District, Harbin 150036, China
[3] School of Engineering Technology, Northeast Forestry University, No. 26, Hexing Road, Xiangfang District, Harbin 150040, China
[*] Correspondence: wxg1972@nefu.edu.cn (X.W.); hhhq1974@126.com (H.H.)

**Abstract:** The simulation analysis numerically investigates the thermoelastic lubricated interfacial Textured Micro-Element (TME) load-bearing contact, a theoretical model is proposed, and the effective friction reduction and Anti-Thermoelastic Scuffing Load bearing (ATSLB) capacity between random rough Meshing Teeth Surfaces (MTS) are presented, the mechanism linking interfacial thermoelastic lubrication, TME meshing friction reduction and ATSLB is revealed. The real contact domain area between MTS with multi-scale Micro-Element Textures (MET) is obtained for the numerical calculation of the three-dimensional equivalent TME contact volume, which is the correlation bridge between friction reduction and ATSLB of the thermoelastic lubrication interface. The proposed theoretical model predicts the time-varying behaviour of the textured meshing interface friction reduction with TME contact load under thermoelastic lubrication conditions. Numerical simulations show that the textured interface meshing volume is the key to solving the load-bearing problem of line contact between randomly rough teeth surfaces. The friction coefficients of the MTS are reduced by 13–24%. The lubricated load-bearing and friction reduction behaviour between the textured MTS is quantified by the thermoelastic voids of TME interface and actual meshing volume ratio, which provides a new perspective for further insight into the lubrication and friction reduction behaviour between the MTS with multi-scale MET-ATSLB coupling mechanism.

**Keywords:** meshing teeth surfaces; multi-scale characteristics; micro-element textures; thermoelastic lubrication; friction reduction; anti-scuffing load-bearing

## 1. Introduction

Meshing teeth surfaces (MTS) texturing by creating multi-scale Textured Micro-Element (TME) features has evolved into a feasible technology to effectively improve the tribological behaviour of gear transmission system components and enhance the lubricated load-bearing capacity [1]. Numerical methods are adopted to model and simulate the lubricated performance by inducing changes in the multi-microscale geometry of the contact interface. The multi-microscale structures perform better in terms of the anti-scuffing load-bearing capacity for the same amount of oil film thickness [2]. The anti-scuffing load-bearing behaviour between elastically lubricated MTS, which is sensitive to the thermal effects of tribological problems, usually occurs in a range of scales from the macroscopic representative dimensions of line contact (e.g., the loaded nominal contact length for thermoelastically lubricated meshing requirements) to the microscopic level. Additionally, multiple scales are included, which are corresponding to the parameter hierarchy of pre-designed configurations (micro-element texturing structures, oil film inhomogeneities, etc.) common at the meshing interface. Numerical approach is based on the deformability of biomaterials to investigate the adaptive surfaces load-bearing properties [3–5]. The research purpose

of this subject is to engineer multi-scale micro-texture distribution patterns into MTS to improve lubricating performance, reduce friction and increase ATSLB capacity.

Previous attention has simply restricted interfacial contact to the classical mechanical level, and it has been noted that even if the MTS are not initially covered with micro-textured topography, the multiple scales are contacted by wear (which is stochastic in this case) or by micro-deformation due to thermoelastic lubrication in a wear-free process, which is just one result of the formation of voids at the contact interface [6]. Realistic line contact at meshing interfaces requires physical phenomena occurring at the rough scale to bridge the discrepancy between the microscopic scale (e.g., the source of frictional heat) and the macroscopic scale (experiencing friction and applying slip motion). In recent years, different results in this respect have been increasingly reflected in a wide range of contact tribology studies related to the mechanisms of lubrication load-bearing at gear meshing interfaces (rolling slip line contact in meshing domain). Numerous reported and researched results indicate that micro-texturing technology does significantly improve the friction reduction properties and load-bearing capacity of the gear meshing contact interfaces. Multi-scale micro-textures configurations can be designed to gear MTS to improve lubrication performance and carefully selected micro-textures are expected to help retain lubricating oil and enhance hydrodynamic effects at the gear meshing interface. Textured MTS are generated based on the TME model and "validated" by numerical simulation.

The concept of micro-textured contact interface is not a new term. Early similar reports date back to 1966 [7], when a performance study of the interface texturing topography of mechanical valves is presented. The research results show that the contact interface with micro-textured topography can weaken friction and reduce wear. A mathematical model of a mechanical seal interface with a hemispherical pore micro-texturing is proposed and used to predict mechanical seal performance as a function of interface pore geometry and contact applied pressure [8–10], which inspires pioneering research of contact tribology in micro-textured contact at the laser interface and is also only an unveiling. Subsequent scholars have conducted experiments and theoretical feasibility studies on mechanical seals with micro-dimples created using laser interface texturing technology [11–13]. Experiments show that the coefficient of contact friction is reduced in all cases of micro-textured interfaces compared to those non-micro-textured interfaces. The friction behaviour of piston-cylinder assembly contact subsets based on micro-texturing morphology with laser interface is studied and experimented. The results show that the interfacial friction coefficient can be reduced by 25–35% with laser micro-textured profiles [14]. Using optical interferometry technology, a steel ball with transverse grooves is used to experimentally simulate the rolling and sliding forms of similar involute gear pairs on a non-metallic plate. The research shows that under Elastohydrodynamic Lubrication (EHL) conditions, the microfilm thickness at the contact interface with transverse grooves micro-textured morphology is obviously thinner [15]. The influence of thermal effects is not considered and the Thermo-Elastic Hydrodynamic Lubrication (TEHL) model is not introduced, which is a fly in the ointment. This study describes the numerical simulation of multiscale micro-textured MTS in TEHL contacts. The focus is on the selection and determination of MTS micro-texturing distribution patterns based on TEHL performance. Contact multi-scale geometry, TME distribution density, interface materials and operating conditions all affect the performance of textured MTS. Evaluating the design rationality of an micro-textured MTS requires extensive numerical simulations. A complete experimental evaluation of micro-textured MTS properties is expensive and time-consuming. Multiscale MET techniques involving micro-textured MTS generation and analytical model-based performance evaluation are considered to be the most effective means to find generalized directions for TME creation and configuration optimization.

In the process of high-speed and heavy-duty gear transmission systems (such as marine power rear transmission systems) operation, problems, such as MTS lubrication performance-friction/wear behaviour-carrying capacity, will inevitably be encountered. At present, there are many technical solutions for solving the above problems. As an effective technical method, micro-textured MTS technology has attracted extensive attention from scholars at home and

abroad. In recent years, the contact interface micro-texture technology has achieved great results in the research of the meshing interface of gear pairs, which is of great significance for improving the lubrication performance and friction wear behaviour and load bearing capacity of MTS at high speed and heavy load. The effects of multi-scale micro-textured distribution patterns, such as dimple geometry shape, dimple size, and dimple density, on the contact tribological properties of interface morphology are investigated and a lubricated column pin-on-disc experiments is presented on non-steel and steel samples [16–18]. These results suggest that the multi-scale distribution patterns of micro-textures, such as dimple size and dimple density, has a profound effect on the contact tribological properties of interface morphology compared to dimple geometry. According to the above report, the authors speculate that the multi-scale depth micro-dimples with micro-texture interface are more effective in increasing the thickness of the lubricating oil film, which is expected to improve the load-bearing capacity of the interfacial oil film in TEHL contact. Based on pin-disk experiments with multi-scale micro-texture geometries (a series of square lattice micropores and parallel TME groove samples), the experimental results illustrated by the Stribeck curves show that in the EHL state, the coefficient of friction at the interface of the squared micro-textured holes is reduced by approximately 6% and the coefficient of friction at the interface of the parallel micro-element textured grooves is increased by approximately 81% in the hybrid TEHL state [19–21]. A theoretical model for studying the laser interface micro-texture topography pattern of non-rigid elastomers in EHL contacts is proposed and the Reynolds equation and the elasticity equation of non-rigid elastomers are solved. These solutions show that the desired reduction in the contact friction coefficient by 30% can be achieved under optimum lubrication conditions at the interface [22]. The copper-steel disks experiment with the interface EHL state is conducted, considering the low-velocity sliding line contact problem, the influence of interface indentation size on the friction coefficient is analysed, and a numerical model for predicting the pressure distribution in the interface indentation is proposed [23]. Related numerical simulations and experimental studies of contact interface micro-texturing have also been addressed in other applications, such as reducing friction in bearings and improving the performance of mechanical transmission systems [24–27]. Fully designed and optimized MTS textured configurations are considered as a promising approach to improve the ATSLB capacity and reduce slip-line contact friction losses. The purpose of this topic is to analyse the local lubrication mechanism of MTS thermoelastic interfacial TME load-bearing contact from a microfluidic perspective, while recognizing the relationship between multi-scale micro-element textures and oil film formation in the entire MTS interaction. For this research aim, a hydrodynamic lubrication model of micro-textured MTS is engineered.

Reported results show that multi-scale parameters, such as the size and morphological density distribution of interfacial micro-dents, perform a non-negligible and unimaginable role in the elevation of the contact load-bearing capacity and reducing the coefficient of interface friction. A finite-difference computational model is proposed to investigate the effect of interfacial micro-texturing with multiscale dimple point geometries on the performance of hydrodynamic journal bearings to assess the load-bearing capacity [28–30]. The tribological properties of micro-textures with annular, transverse and longitudinal indentations at the steel rings interface machined with a laser are investigated. Observations reveal that the depth behaviour of multiscale indentations micro-textures is highly dependent on the relative sliding velocity [31–33]. The search for the desired optimum interfacial contact properties (especially similar to involute gear pair linear contact) can be pre-designed for interfacial TME structures or customized with the help of chemical modifications (simulating rolling/sliding contact with micro-textured interfaces), as has been described by our group in contact interface theory in recent years. The results of years of research by scholars' report that the lower the roughness of the meshing interface, the better the line contact performance. In contrast, the micro-textured MTS with multi-scale geometry exhibits the expected lubrication performance, frictional wear behaviour and load bearing capacity.

In this paper, a method for the approximate homogenization of oil film lubrication with micro-textured effective contact interfaces is proposed and the approximate homogeneous Reynolds flow dynamics problem caused by interfacial micro-textures is described, the multi-scale properties of the micro-textures (including micro-texture shape, depth and micro-elements morphological density distribution) are determined and local slip length variations in MTS line contact caused by TEHL viscous resistance textures at the micro-elements scale are considered. This provides a step forward in understanding subsequent problems, focusing in particular on the microscopic meshing conditions derived for line contact gear pairs of heavy-duty transmission systems with interfacial micro-textures. An optimal solution strategy is proposed to investigate the microtextural properties of the micro-texturing interface under TEHL contact to modulate the multi-scale MET morphology to maximize the ATSLB (see Figure 1). This topic analyses the latest background and recent trends of the micro-texture of the contact interface, emphatically pointing out the purpose and development status of this research. Lubrication performance, frictional wear behaviour and load bearing capacity mechanism of multi-scale micro-textured MTS is summarized, and the reason for the formation of the mechanism is explained. The TME configuration optimization of contact morphology and interface parameters is discussed and the optimal micro-texture parameter range of MTS is obtained. The numerical generation process and the hybrid TEHL model for 'microfabricated' MTS constitute a virtual analysis system, which is used to create, study and compare a series of textured MTS, and perform numerical experiment simulation to verify and evaluate the TEHL performance of the generated MTS. Practical engineering applications of MTS texturing in gear transmission system to reduce friction, control wear, improve lubrication and enhance ATSLB capability are attracting increasing interest. It is accepted that the optimization of MTS textures should be tailored to the specific requirements of the application.

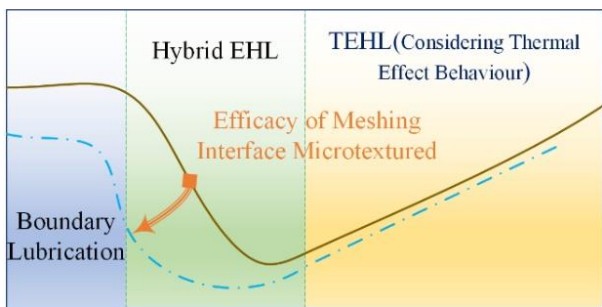

**Figure 1.** Discrete and continuous micro-element textures.

## 2. Micro-Textured Interface Thermoelastic Hydrodynamics Homogenization Model

Micro-texturing technology does significantly improve the friction reduction properties and load-bearing capacity of the contact interfaces [34–36]. The multi-scale geometric parameters of micro-textures are one of the core factors to optimize the microfabricated interfaces benefits. Emphasis on these known results provides a significant improvement in the general description of friction-wear and lubrication-load-bearing. The micro-textured contact interface is presented in a pre-designed engineering. Needless to say, the correlation and mechanistic understanding of each complexity parameter has not been fully explored, the analysis describes only a limited number of hypothetical cases that fail to break through most of the known mechanisms and is limited to crater area ratios and pattern shapes of discrete micro-element textures. Furthermore, there is little pre-research work on multi-scale contact interface micro-textures from the perspective of geometric parameter characterization.

Micro-textured contact interface forms are generally divided into two types, namely, continuous and discrete (see Figure 2). Multi-Scale features of micro-element geometric shapes, such as circles, rectangles, triangles, honeycombs, etc., are promoted to discrete micro-textures. The continuous micro-textures in this model are in the form of parallel

straight lines or arrays of crossed curves. These microelement-scale characteristic differences in micro-textured interface design will alter the contact pressure profile within the gear meshing domain. The technical challenge of micro-texturing is to determine the critical dimensional relationship between the distance (pitch, array, orientation angle) and the control of the gear contact edge profile between the width, length and depth of interfacial micro-scale features (see Figure 3). The search for correlation between key optimal micro-element shape design parameters and geometric scale parameters is necessary to obtain maximum frictional wear reduction benefits and thus improve performance and pursue this concept for lubricated load-bearing engineering applications at gear meshing interfaces. In this study, pre-designed micro-element texturing patterns of slip line contact interfaces with Electrohydrodynamic Lubrication (EHL) conditions considering thermal effect behaviour reveal the correlation between friction and load-bearing capacity of some core parameters, such as depth and width under hybrid Thermoelastic Hydrodynamic Lubrication (TEHL). As microscopic geometry emerges as insights into mechanisms of anomalous behaviour, scholars have explored further insight into the mechanisms underlying the lubrication-load bearing association of MTS micro-textures.

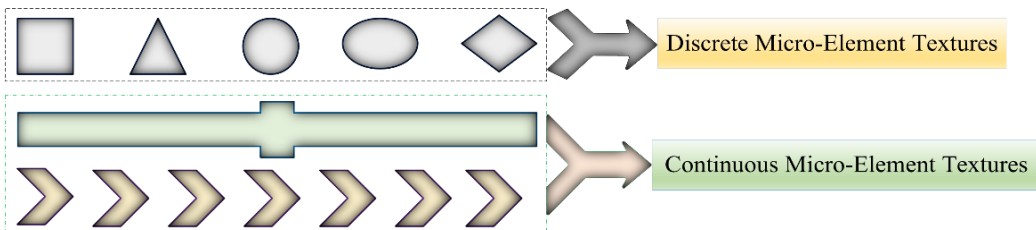

**Figure 2.** Discrete and continuous micro-element textures.

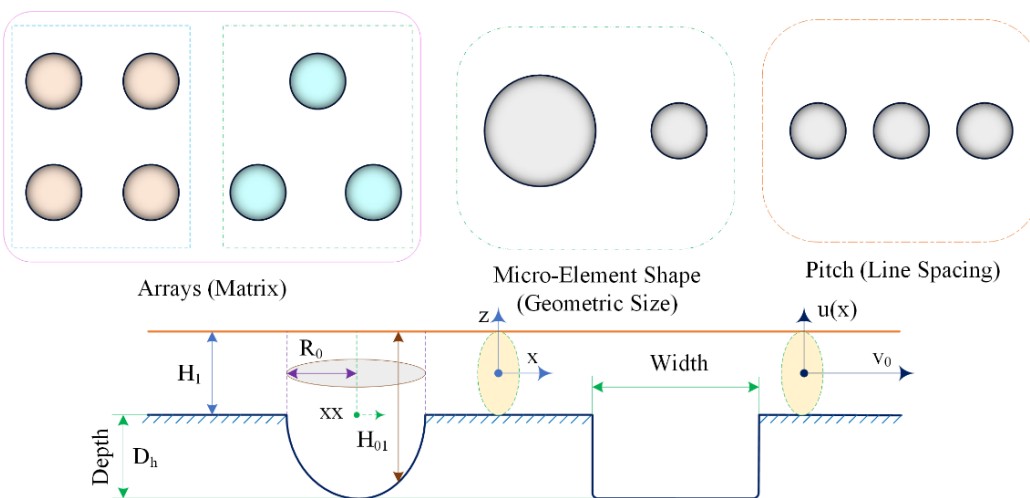

**Figure 3.** Multiscale characterization for micro-element textures of contact interfaces (Distribution density and geometric parameters).

The TEHL region of the effective cross-section of the column pin and the disc is simulated as a linear contact of the gear pair meshing interface, where the Reynolds equation is the master governing equation for dynamic pressure lubrication flow between the MTS contact interfaces. The dimensionless expression of the Reynolds equation [37] is written:

$$\frac{\partial}{\partial X}\left(\psi \frac{\partial P}{\partial X}\right) - \left(\frac{\partial(\rho_m H)}{\partial X}\right) = 0 \tag{1}$$

where the parameters are described as:

$$X = x/a, \psi = \rho_m \left( H_c R_x/a^2 \right)^3 / \left( 12 \mu_m \eta_0 R_x^2 / P_{\max} a^2 \right) \eta_m, U = \mu_m \eta_0 / E_1 R_x \quad (2)$$

Here, $a$ denotes half-width of the MTS contact region, $P_{\max}$ represents the maximum Hertzian pressure P, $H_c$ is the central oil film thicknesses H, $\rho_m$ is lubricating oil density at Hertzian pressure P, $\mu_m$ is the friction coefficient mean value, $\eta_0$ is the viscosity of lubricating oil, $\eta_m$ is the average viscosity of lubricating oil and $E_1$ is the equivalent elastic modulus. The film thickness is calculated from the mean gap after deformation, flow and roughness contact are treated in a unified model. In the hydrodynamic lubrication region, the pressure is controlled by the Reynolds equation expressed as follows:

$$\frac{\partial}{\partial x} \left( \frac{\rho_m H_c^3}{12\eta_m} \frac{\partial P}{\partial x} \right) + \frac{\partial}{\partial y} \left( \frac{\rho_m H_c^3}{12\eta_m} \frac{\partial P}{\partial y} \right) = U \frac{\partial(\rho_m H_c)}{\partial x} + \frac{\partial(\rho_m H_c)}{\partial t} \quad (3)$$

where the $x$ coordinate coincides with the direction of motion and $U = \mu_m \eta_0 / E_1 R_x$.

In this topic, a non-finite length cylindrical line contact form is assumed. A segment of the TME computational domain for solving the TEHL fluid dynamics overlaps along the x-line direction (see Figure 4), where $d$ indicates the depth (height) of the TME and $l/a$ denotes the aspect ratio of a single TME. In the simulation analysis of TEHL line contact, the approximation errors caused by considering the classical asymptotic assumptions are obvious and non-negligible. Coupled with the differences in multiscale properties, it is difficult to achieve, to find more accurate numerical solutions from a complex set of equations. Assuming that multi-scale textured interfaces are microhomogeneous and periodic, a formal approach to decoupling macroscopic and microscopic scales is adopted, and a homogenized micro-TEHL contact model for interfaces with TME is proposed, which considers the non-negligible deformation caused by compressive load-bearing and thermal effects at the microscopic scale, which extends the general applicability of classical progressive homogenizations methods.

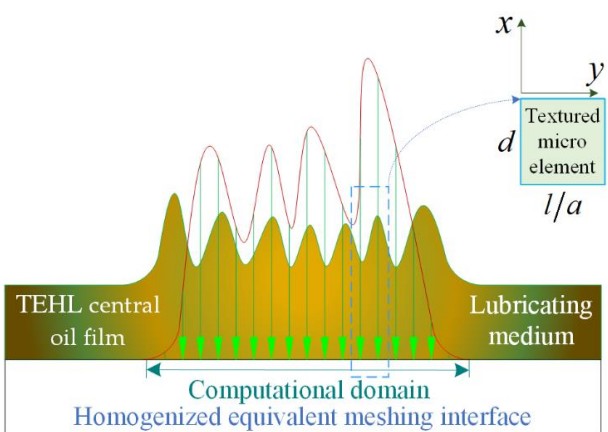

**Figure 4.** A Homogenized equivalent meshing interface section of TME computational domain.

The dimensionless expression of lubricating oil film thickness [38] is identified as:

$$H_{OFT} = H_1 + \frac{x^2}{2a^2} + \zeta \quad (4)$$

where $\zeta$ is the thermoelastic deformation (the elastic deformation considering thermal effects), which is:

$$\zeta = \left[ -\frac{((1-\nu_1^2)/E_1 + (1-\nu_2^2)/E_2 + \cdots (1-\nu_n^2)/E_n)}{\pi} \right] \int_{x_a}^{x_{a'}} P(x) \ln \left( \frac{x}{R_0} - \frac{x'}{R_0} \right)^2 dx \quad (5)$$

Previous studies in the subject group have reported that the convergence of the teeth meshing phase slows down as the meshing thermal expansion stiffness factor increases. This suggests that the meshing thermal expansion stiffness factor is not effective in suppressing contact shock, but improves the instability of the MTS line contact lubrication performance, frictional wear behaviour and load bearing capacity.

Seeking a micro-textured indentation that optimally characterizes the asperity geometry with the concave-convex interface, the textured indentation has an aspect ratio (individual dimple), $\kappa$, and a dimensionless minimum spacing between the gear meshing interfaces, $d_s$, which is specified below:

$$\begin{cases} \kappa = \dfrac{D_h}{2R_0} \\[2mm] d_s = \dfrac{H_c}{2R_0} \end{cases} \tag{6}$$

where $x$ is defined by the geometric centre of a micro-element dimple with radius $R_0$ and depth $D_h$, see Figures 3 and 4. Further exploring the contribution of micro-element dimples with the oil film thickness relationship of gear meshing interfaces of Equation (4) is revised to the form of Equation (7), which is written as:

$$\frac{H_{01}(x)}{H_1} = \begin{cases} 1 + \dfrac{\sqrt{\left(\dfrac{1+4\kappa^2}{2\kappa}\right)^2 - x^2} + \left(\kappa - \dfrac{1}{4\kappa}\right)}{2d_s}, (x \leq 1) \\[4mm] 1, (x > 1) \end{cases} \tag{7}$$

Lubricating oil properties (e.g., viscosity and density parameters indicators) between gear meshing interfaces vary with temperature and pressure. The Roelands characterization Equation (8) is introduced to correct for viscosity as a function of temperature-pressure, which approaching exponential values of high average values [39].

$$\frac{\eta}{\eta_0} = -\exp\left[(9.67 + \ln\eta_0)\left(1 - (5.1 \times 10^{-9}P + 1)^{0.68}\right)\right] \tag{8}$$

Dowson and Higginson derived the relationship between the indicators of density and pressure [40], which is:

$$\frac{\rho}{\rho_0} = \left(\frac{0.6P}{1 + 1.7P} + 1\right) \tag{9}$$

In summary, the load-bearing balance equation in the textured meshing interface line contact problem is described as the integral of the pressure multiplied by the microelements contact area in the gear meshing domain equals the total amount of the applied loads [41], which is expressed in dimensionless form as:

$$\varpi = \int_{-\infty}^{+\infty} P\,dx \tag{10}$$

where $\varpi$ is used as an indicator parameter for load-bearing capacity.

In this paper, the meshed load-bearing characteristics of meshing TME interfaces in a hybrid TEHL state are simulated on a column pin-disc test rig, the load-bearing capacity of the homogenized oil film with the meshing TME interface is described and the enhanced homogeneous shared tribological performance of the TEHL oil film with meshing TME interfaces load-bearing capacity are predicted for optimal assessment.

In a hybrid TEHL regime, the overall load $F_O$ applied on the Meshing Teeth Surfaces (MTS) is shared by the interfacial contact between the TEHL oil film $F_{fh}$ and the TMEs $F_{CI}$, which is expressed as follows [42]:

$$F_O = F_{fh} + F_{CI} \tag{11}$$

By multiplying both sides of Equation (11) by $1/F_O$, two setting constants $c_1$ and $c_2$ are introduced to describe the contribution of the shared loads enhanced by the non-textured and textured gear meshing interfaces, respectively, under TEHL.

$$\frac{F_O}{F_O} = \frac{F_{fh}}{F_O} + \frac{F_{CI}}{F_O} = \frac{1}{c_1} + \frac{1}{c_2} = 1 \tag{12}$$

The friction force generated by TEHL oil film shearing between gear meshing interfaces and the friction force generated by TME contact are similarly described [43].

$$F_{ff} = F_{fTEHL} + F_{fTME} \tag{13}$$

where $F_{ff}$ indicates the friction force, $F_{fTEHL}$ is the friction force of TEHL oil film shearing and $F_{fTME}$ denotes the friction force of TME contact. The friction coefficient mean value $\mu_m$ is defined as follows, which is:

$$\mu_m = \frac{F_{ff}}{F_0} \tag{14}$$

The algorithm for solving the friction coefficient of the meshing interface starts with the initial values of the assumed constant coefficients $c_1$ and $c_2$. The load-bearing capacity of the TEHL oil film with a TME interface (micro-concave peaks) is pre-determined to be known and the TEHL oil film thickness in the hybrid lubrication condition is evaluated by considering the transient contact process of purely elastic to thermoelastic deformation of the textured interface along the meshing line and the loads carried by each TME in the meshing domain is calculated [44], which is as follows:

$$F_j = \begin{cases} F_{jk} = 0.75 E' R_m^{1/2} \omega_j^{3/2} \\[2mm] F_{jkl} = \left(1 - \dfrac{3(\ln \omega_j - \ln \omega_k)}{5(\ln \omega_l - \ln \omega_k)}\right) G_H A_{jkl} \\[2mm] F_{jl} = 2\pi R_m G_H \omega_j \end{cases} \tag{15}$$

where $F_j$ denotes the load carried by TME $j$, $F_{jk}$ is the load carried by a single radius of the TME (indentation), $F_{jkl}$ is the load carried by the mean radius of the micro-concave peak tip, $F_{jl}$ is the load carried by the contact area of the TME interface $E'$ identifies the equivalent elastic modulus, $R_m$ is the mean radius of the micro-concave peak tip, $\omega_j$ is the Textured Micro-Element (TME) (indentation) of TME $j$, $\omega_k$ is the TME (indentation) of TME $k$, $\omega_l$ is the TME (indentation) of TME $l$, $A_{jkl}$ is the interface contact area of a TME in the TEHL regime and $G_H$ is the gear material hardness. The total forces subjects to the calculation of the load carried by the TME, $F_{fTME}$. The value of $F_{fTME}$ is compared with $F_{ff}/c_2$ to determine if the initial assumption of a constant coefficient holds. If this does not hold, another constant factor is assumed and the determination continues until convergence is achieved and the friction factor is evaluated. The coefficient of friction involves two component terms: an interface term due to the TME (micro-concave peaks) contact and a load-bearing term due to the friction in the TEHL oil film.

In this research topic, the Newton-Raphson method is used to solve the Reynolds equation. This is a faster shortcut than the traditional direct method. Another surprising advantage of this approach is better convergence at high load line contact. A brief description of the simulation process is as follows. The TME discretization equation is controlled by the finite difference method, the initial Hertzian pressure distribution of the textured

MTS is preset, the estimated value of the TEHL oil film thickness between the meshing interfaces is sought and the optimal value of the minimum oil film thickness is solved, which ensures that the load balance equation applying the texturing interface contact force and the micro-element homogenization between the TEHL is updated in real time at the minimum film thickness while solving all equations to predict the contact pressure distribution and interface oil film thickness variation. Algorithm flow chart of multi-scale numerical model algorithm for lubricated MTS with TME considering ATSLB capacity is indicated in Figure 5. The simulated data predicting the coefficient of friction are obtained at different speeds. The expected result is that the friction decreases with increasing speed for increasing lubricant film thickness. The simulated coefficient of friction is compared with the variation in speed in the presence of a textured MTS. The simulated friction coefficients are obtained at different speeds. The numerically calculated best textured samples show a better agreement.

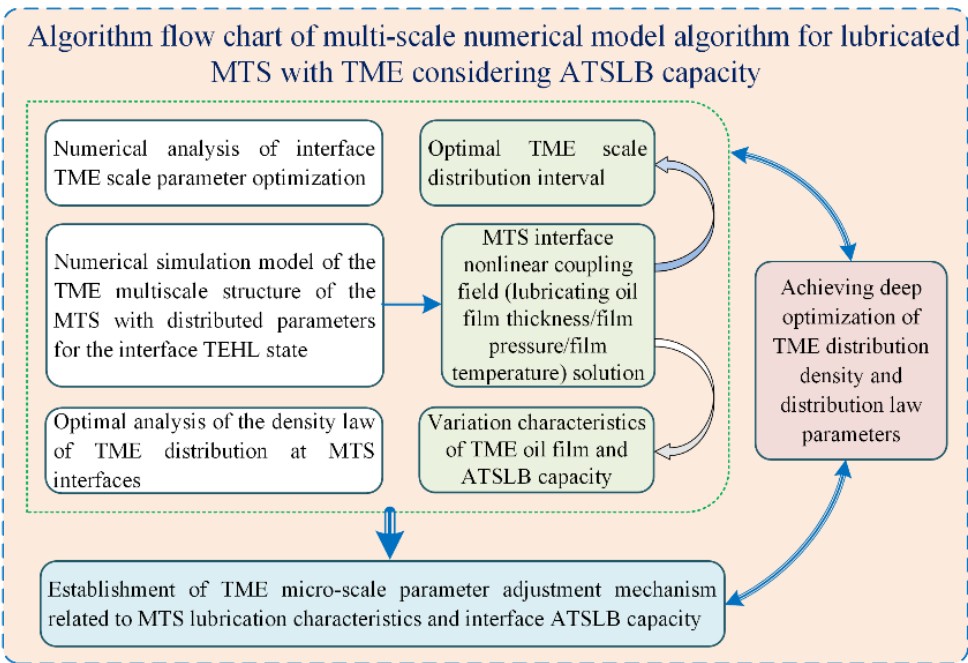

**Figure 5.** An algorithm flow chart of multi-scale numerical model algorithm for lubricated MTS with TME.

### 3. Simulation of MTS Contact Experiments with TME Characteristics

A column pin-disc test rig is presented to experimentally simulate the correlation effect of different TME contact parameters on the load-bearing properties and textured meshing interface under TEHL conditions. The TME interface contact experiments are performed on a test rig with a column pin-disc geometry, which is illustrated in Figure 6. The MTS line contact model for involute spur gears is experimentally simulated using TME characteristic parameters, which are perpendicular to a selected micro-texture interface along the meshing line slip direction, which is periodically and uniformly distributed in the meshing domain.

The column pin is made of 40 Cr with a diameter of $\phi$20 mm and a functional disc surface of $\phi$19.3 $\pm$ 0.2 mm is formed by grinding. The circular meshing domain surfaces are implanted by means of laser micromachining. The width of the micro-texture is determined to be 50 μm, 100 μm and 200 μm, respectively, the depth is 30 μm and the coverage rate is 75%. Simulated meshing loads and friction behaviour data acquisition is performed by means of an electronic force sensor. A lead port is provided at a distance of 0.50 mm below the centre of the functional disc surface for the uninterrupted detection of the MTS contact temperature with a TME by an embedded thermocouple sensor. The simulated

experiments are performed with set loads of 100 N and 200 N and slip linear velocities of 1.0 m/s and 2.0 m/s. During the entire cycle time (meshing cycle) of 180 s, CD40 lubricant is continuously delivered from the input to the column pin-disc interface contact area (gear pair meshing domain).

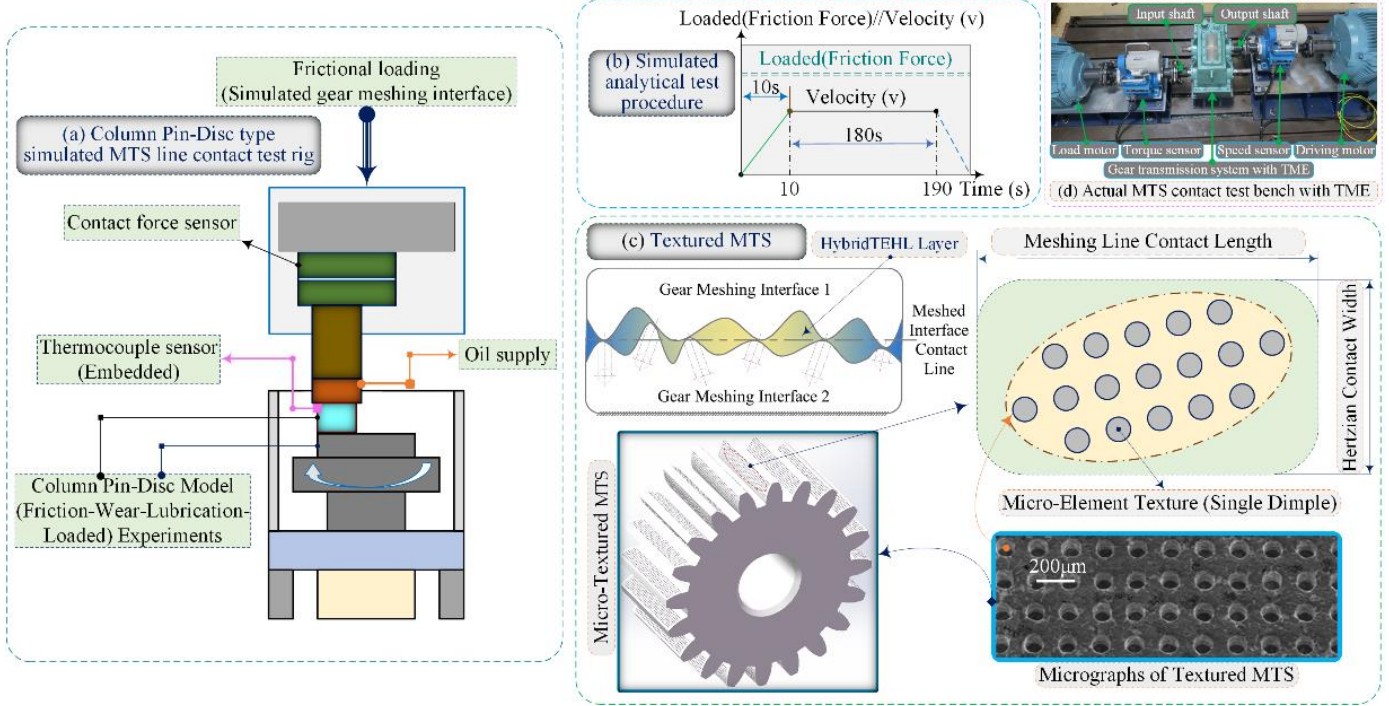

**Figure 6.** Experimental details of simulated micro-elements texturing MTS under TEHL: (**a**) Column Pin-Disc type simulated MTS line contact test rig, (**b**) Simulated analytical test procedure, (**c**) Determination of thermocouple sensor (Embedded) position (for measuring the contact temperature of MTS between lubricating interfaces) and (**d**) Actual MTS contact test bench with TME.

## 4. Results and Discussion

The results of the numerical calculations and experimental simulation studies are reported and discussed in the following presentations. The numerical calculation and simulation data of the experimental parameters are presented in Table 1. Herein, the input parameters include TME pattern and geometry and the output parameters include TME interface pressure distribution, TEHL oil film thickness and oil film temperature.

**Table 1.** Data parameters used in numerical simulation experiment.

| State | Interface TME Aspect Ratio | Contact Force Applied to MTS (N) | MTS Slip Linear Velocity (m/s) |
|---|---|---|---|
| | - | 100 | 1.0 |
| Un-MTE | - | 150 | 1.5 |
| | - | 200 | 2.0 |
| | 0.20 | 100 | 1.0 |
| TME | 0.15 | 150 | 1.5 |
| | 0.10 | 200 | 2.0 |

The contact pressure distribution and TEHL film thickness variation patterns for MTS without TME characteristics are considered for interface bearing loads of 100 N and 200 N, respectively. The results denote that the minimum load is applied to the interface at a slip linear velocity of 1.0 m/s and the maximum load is applied to the interface at a slip linear velocity of 2.0 m/s. It is difficult for the meshing interface to define an obvious contact

pressure peak at a lower bearing load, and the effect of thermoelastic deformation on the meshing interface deserves high attention in future research, especially for high-speed, heavy-duty MTS. As illustrated in Figure 7.

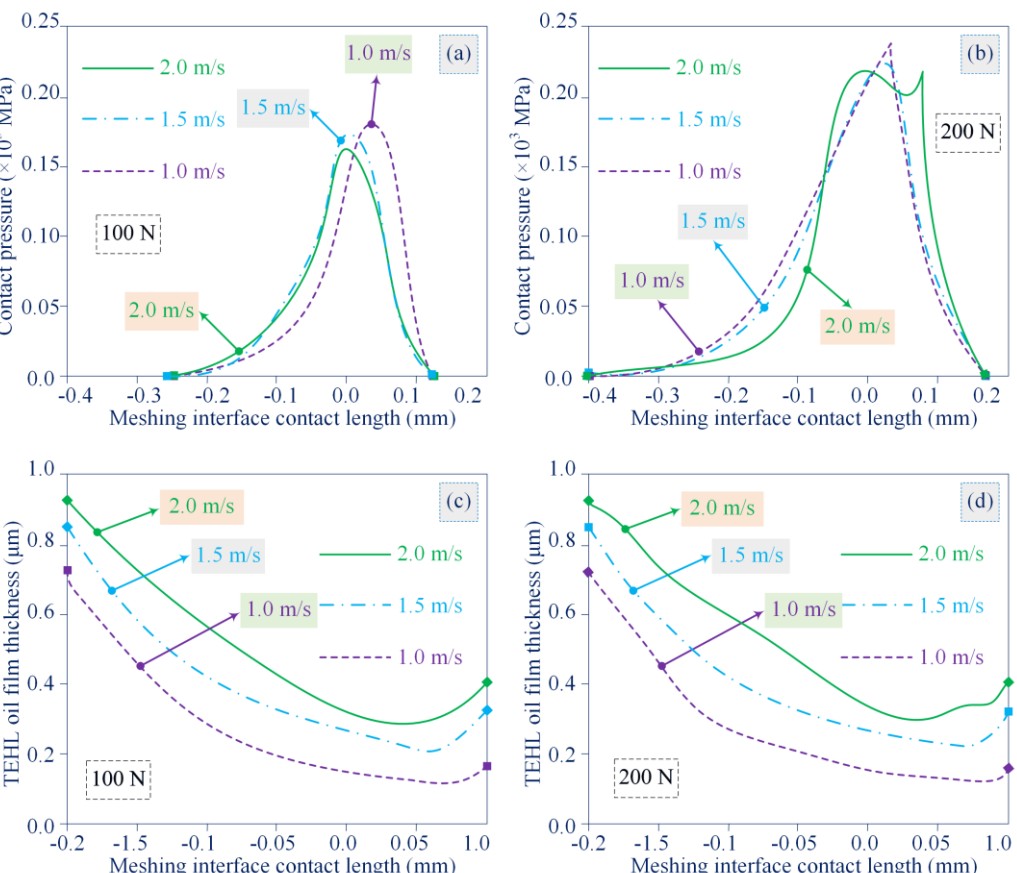

**Figure 7.** Contact pressure distribution and TEHL film thickness variation of MTS without TME characteristics: (**a**) Contact pressure of untextured MTS under a load of 100N varies with linear velocity, (**b**) Contact pressure of untextured MTS under a load of 200N varies with linear velocity, (**c**) TEHL oil film thickness of untextured MTS under a load of 100N varies with linear velocity, and (**d**) TEHL oil film thickness of untextured MTS under a load of 200N varies with linear velocity.

It is deduced from the simulation analysis that increasing the slip linear velocity leads to easier formation of a thicker TEHL lubrication film, which in turn affects the distribution position and amplitude of the contact pressure peak, as presented in Figure 8.

The pressure distribution and TEHL oil film thickness of the micro-textured meshing interface are elucidated for the same applied load and slip linear velocity. The presence of the TME characteristics produces two contact pressure peaks at its edges and a flat and relatively low-pressure contact region in the middle of the meshing area. Note that the decrease in TEHL oil film thickness corresponding to the contact pressure peak is expected to be even lower than the minimum TEHL oil film thickness for an untextured meshing interface. A proven feasibility solution is to increase the average oil film thickness of the MTS by the technical means of micro-textured contact interface. An interesting finding is that the minimum TEHL oil film thickness of the MTS occurs at the leading edge of the TME boundary.

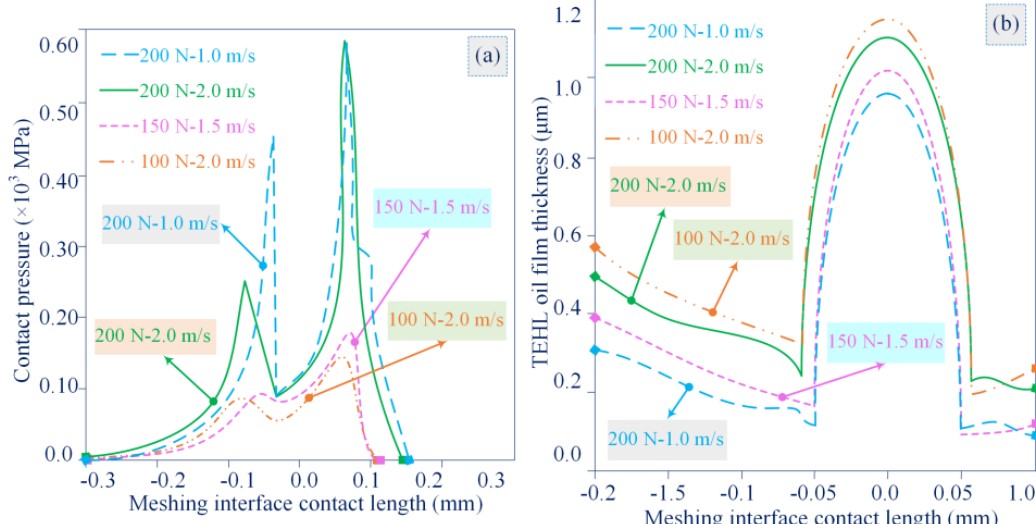

**Figure 8.** Variations of contact pressure and TEHL oil film thickness with textured MTS under 100 N, 150 N and 200 N loads and 1.0 m/s, 1.5 m/s and 2.0 m/s linear velocities: (**a**) Contact pressure with textured MTS under 100 N, 150 N and 200 N loads and 1.0 m/s, 1.5 m/s and 2.0 m/s linear velocities, (**b**) TEHL oil film thickness with textured MTS under 100 N, 150 N and 200 N loads and 1.0 m/s, 1.5 m/s and 2.0 m/s linear velocities.

In Figure 9, under the applied load of 200 N and slip linear velocity of 2.0 m/s, the effect of TME geometry on the contact pressure distribution and TEHL oil film thickness is detailed. The behaviour of three TME depths of 5.0 μm, 10 μm and 20 μm corresponding to different interface TME aspect ratios of 0.1, 0.15 and 0.2 are analysed and compared. One of the most desired goals of MTS micro-texturing is to improve the load-bearing capacity of the contact interface. The TEHL average oil film on MTS with TME characteristics is thicker than that of non-textured MTS and the contact pressure is distributed in a larger meshing area. Moreover, two contact pressure peaks are formed at the edge of the TME. When the interfacial TME aspect ratio is not high, the contact area with meshing constant regime is wider and a larger contact pressure peak occurs at the trailing edge of TME to meet the load bearing balance. This larger contact pressure peak causes the TEHL oil film thickness to drop to a smaller magnitude than the minimum TEHL oil film thickness for an untextured MTS under the same conditions.

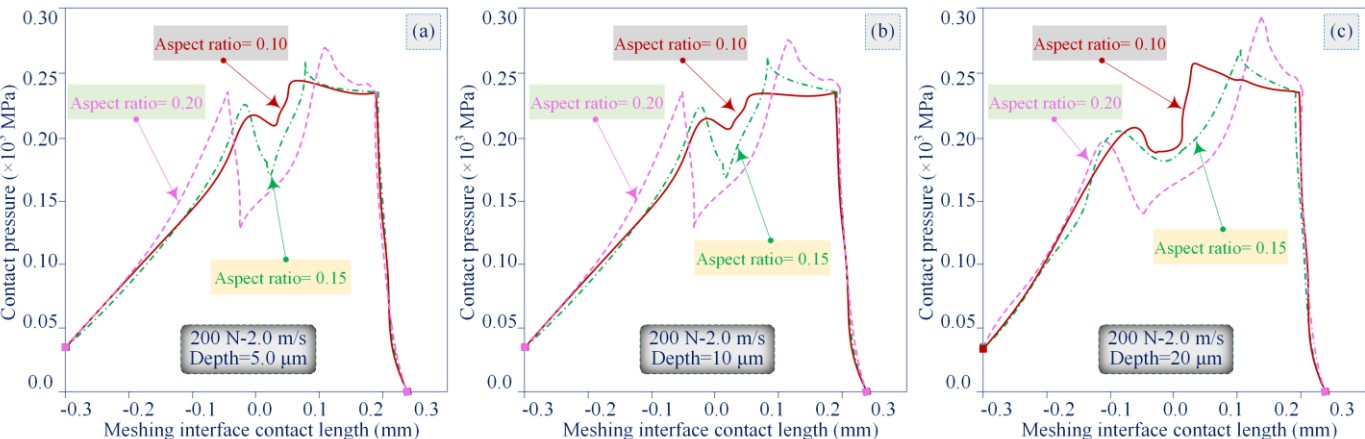

**Figure 9.** Contact pressure variations of interface TME aspect ratios and TME depths under 200 N load and 2.0 m/s slip linear velocity: (**a**) TME depth of 5.0 μm, (**b**) TME depth of 10 μm and (**c**) TME depth of 20 μm.

The minimum TEHL oil film thickness and the average oil film thickness of the microtextured MTS under the same applied load and slip linear velocity conditions are discussed in detail in Figure 10.

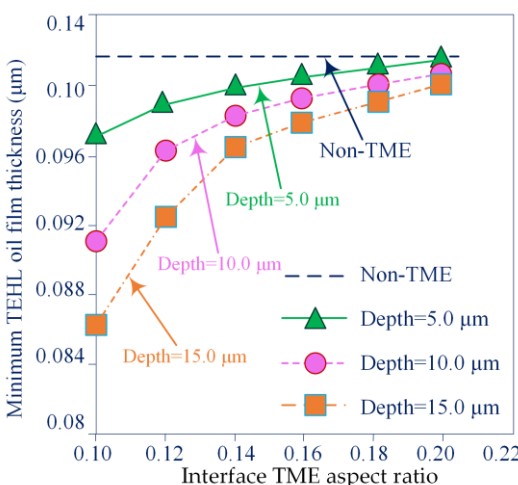

**Figure 10.** Variation curves of minimum TEHL oil film thickness for different TME depth scale sizes with different interface TME aspect ratios at a load of 200 N and a slip linear velocity of 2.0 m/s.

The analysis results show that increasing the interface TME aspect ratio leads to an increase in the minimum TEHL oil film thickness. In this case, the minimum TEHL oil film thickness of non-micro-textured MTS is larger than that of micro-textured MTS with different TME depth values. Increasing the interfacial TME aspect ratio leads to a decrease in the average oil film thickness of the MTS. The average oil film thickness without micro-texturing is smaller than that on micro-textured MTS. The average film thickness is calculated on the entire contact area and the minimum TEHL oil film thickness is one of the key parameters to determine the tribological performance of micro-textured MTS.

Asperities with a height greater than the TME minimum TEHL oil film thickness will generate greater friction, heat, and wear particles even though the height of these asperities may be less than the MTS average oil film thickness.

Figure 11 illustrates the characteristic function relationship between the maximum load-bearing contact pressure variation and TME geometry of MTS with different microtextures and non-micro-textures. The results show that the loaded peak contact pressure of MTS with different micro-textures is higher than that of non-micro-textured MTS, especially for TME with not high interface aspect ratio. The lower interface TME aspect ratio makes the TME edge domain wider and the load-bearing contact pressure distribution profile becomes more uniform, the homogeneous balance of the MTS applied loads is ensured as far as possible by a higher loaded peak contact pressure. A further thought-provoking fact revealed in Figure 11 is that the scale variation of the maximum MTS load-bearing contact pressure with TME depth in the TEHL steady state does not fully depict that a meshing interface with a deeper scale TME necessarily leads to better MTS tribological properties.

The variation curves in MTS contact pressure for different interface TME depth scale sizes for applied loads are investigated in Figure 12. Table 2 illustrates the numerically simulated areas under the pressure profile for the MTS load-bearing capacity. The results show that the shallower interface TME characteristics are more effective for the MTS load-bearing capacity. An optimally configured interface TME feature emerges between these values. It is observed that the applied load MTS increases almost as the density of the interface TME contact area distribution increases, which indicates that the multi-scale feature structure has a large component over the local interface TME in terms of load-bearing capacity for MTS with interface TME geometry.

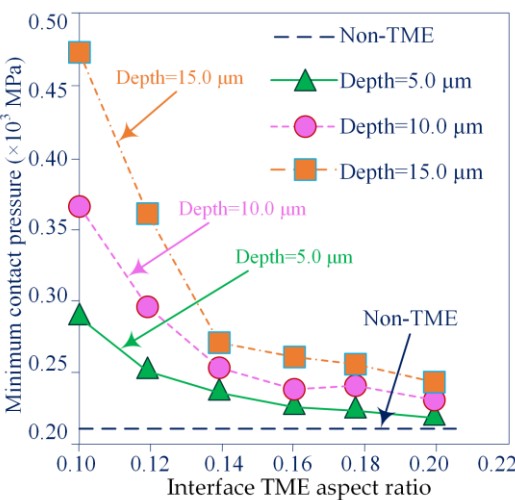

**Figure 11.** Variation curves of MTS maximum contact pressure with interface TME aspect ratio for different interface TME depths.

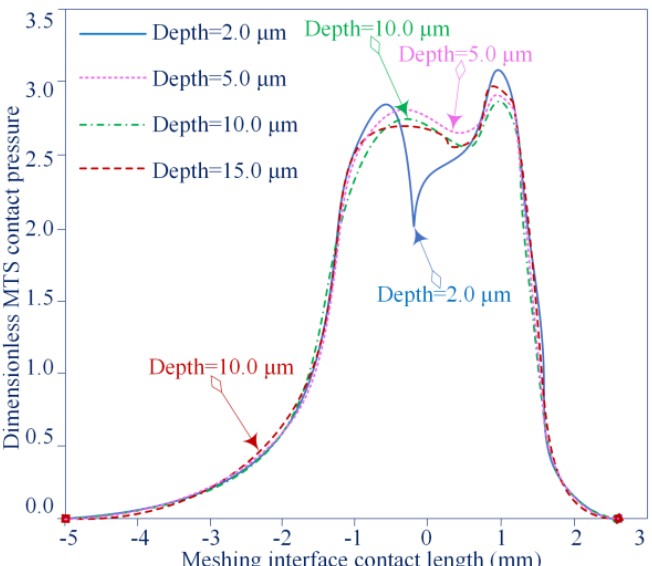

**Figure 12.** Variation curves in MTS contact pressure for different interface TME depth scale sizes for applied loads.

**Table 2.** Correlation effect of interface TME depth scale on MTS load-bearing capacity.

| Interface TME Depth Scale | Dimensionless MTS Load-Bearing Capacity |
|---|---|
| 2.0 μm | 9.17 |
| 5.0 μm | 9.12 |
| 10.0 μm | 9.21 |
| 15.0 μm | 9.10 |

Through this work, the homogenized micro hydrodynamics of the MTS with interfacial TME characteristics in the TEHL steady state is approximated accurately, the microgeometry of the multiscale structure with contact interface TME features is simulated, the existence of an optimal geometry that is particularly effective for the load-bearing capacity of the MTS is proposed and discussed and the validity of the optimized MTS with interfacial TME features significantly affecting the load-bearing capacity of the macroscopic properties of the generic slip linear contact is confirmed.

## 5. Conclusions

The correlation effect of MTS micro-texturing on the tribological performance of TEHL slip linear contact is investigated through simulations and numerical computational modelling to analyse and predict the load-bearing capacity of MTS with and without TME characteristics, which is used to evaluate the friction coefficient of micro-textured MTS. Mating interfaces based on simulated micro-textured MTS lubrication performance-friction/wear behaviour-ATSLB capability concentrated conformal slip wire contact are investigated using TEHL formulations.

The slip linear velocity performs an important role in the formation of TEHL oil film thickness and MTS load-bearing capacity. Under sliding contact with low linear velocity, the TEHL oil film thickness decreases due to the existence of interface TME features, which weakens the MTS load-bearing capacity. With the increase in the interfacial slip linear velocity, the TME multiscale feature presents an interesting performance on the TEHL oil film thickness and MTS load-bearing capacity.

Results are reported from operational data from a number of TEHL steady-state contact experiments using a column pin and disc test rig to simulate the study of the multi-scale structural effects of an MTS with TME characteristics. The results show that the MTS with TME characteristics has an increased load-bearing capacity. Under the lower applied load and higher sliding linear velocity of MTS, the characteristic advantages of texturing technology are more prominent. These interfacial TME features, if the MTS has an appropriate micro-textured multiscale geometry, which significantly improves the thermoelastic lubrication behaviour and load-bearing capacity and the friction coefficient is greatly reduced by 13–24% compared to MTS without interfacial TME features.

A focus of future continuous research work is to conduct experimental research on the dynamic behaviour of gear MTS with various contact interface micro-texture morphologies, so as to deepen the verification of related theoretical findings through the research of this sub-project.

**Author Contributions:** Conceptualization, X.W. and Y.W.; methodology, H.H.; software, J.R.; validation, Y.W., J.S. and J.R.; formal analysis, X.W.; investigation, H.H.; resources, Y.W.; data curation, J.S.; writing—original draft preparation, H.H. and X.W.; writing—review and editing, X.W., Y.W. and H.H; visualization, J.S.; supervision, J.R. and J.S.; project administration, Y.W. and J.S. All authors have read and agreed to the published version of the manuscript.

**Funding:** The research subject was supported by the Doctoral Research Startup Foundation Project of Heilongjiang Institute of Technology (Grant No. 2020BJ06, Yongmei Wang, HLJIT), the Natural Science Foundation Project of Heilongjiang Province (Grant No. LH2019E114, Baixue Fu, HLJIT), the Basic Scientific Research Business Expenses (Innovation Team Category) Project of Heilongjiang Institute of Engineering (Grant No. 2020CX02, Baixue Fu, HLJIT), the Special Project for Double First-Class-Cultivation of Innovative Talents (Grant No. 000/41113102, Jiafu Ruan, NEFU), the Special Scientific Research Funds for Forest Non-profit Industry (Grant No. 201504508), the Youth Science Fund of Heilongjiang Institute of Technology (Grant No. 2015QJ02) and the Fundamental Research Funds for the Central Universities (Grant No. 2572016CB15).

**Data Availability Statement:** Not applicable.

**Acknowledgments:** The authors thank the Huaqiao University, Northeast Forestry University (NEFU) and the Heilongjiang Institute of Technology (HLJIT) for their support.

**Conflicts of Interest:** No conflict of interest exits in the submission of this manuscript, and manuscript is approved by all authors for publication. We would like to declare on behalf of our co-authors that the work described was original research that has not been published previously, and not under consideration for publication elsewhere, in whole or in part. All the authors listed have approved the manuscript that is enclosed.

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
