# Peer review of "Numerical Analysis of Friction Reduction and ATSLB Capacity of Lubricated MTS with Textured Micro-Elements"

_lubricants, doi:10.3390/lubricants11020078_

Round 1
Reviewer 1 Report
Please see the attached review file.

Author Response
January 19, 2023
Dear editor-in-chief
The open access journal Lubricants
We have submitted a research article hoping to be published in the journal Lubricants, titled “Numerical Analysis of Friction Reduction and ATSLB Capacity of Lubricated MTS with Textured Micro-Elements.” The paper was coauthored by XiguiWang, Hui Huang, Jingyu Song, Yongmei Wang, and Jiafu Ruan. We have checked the manuscript and revised it according to the comments. Overall, the comments have been fair, encouraging and constructive. We have learned much from it. We submit here the revised manuscript to meet the evaluation conditions and requirements of the reviewers.
Response to Reviewers' Comments:
Reviewer1:
On behalf of all members of our team, I would like to thank the Reviewer 1 for approving and accepting this manuscript. The authors have once again fully reviewed and refined this revised manuscript, and reflected the refined content in this article. In the long-term in the future, our research team will submit more excellent and high-quality articles to your journal.
Q1. First of all, the introductory section, especially its second half, needs a complete overhaul, as it is without a transparent train of thought regarding the motivation and standalone features of the manuscript. Publications from the literature are listed with their general findings, but it is unclear, how the state of the art in the field of interest evolved over time and what is the current knowledge gap, the manuscript aims to close. Moreover, when just reading the references list, one might get the impression that the research field only started just 5 years ago (as all references are from 2017 or later). That is obviously not the case – as the authors acknowledge themselves, stating that the concepts dates back to 1966 (without offering the original source) – and, accordingly, several important contributions in the field are missing (which is one of the main reasons for the lack of a transparent train of thought). Finally, there is, at least, one strange reference ([27]), which doesn’t seem to be related to the topic at all.
Sincerely respond to Q1 raised by Reviewer 1:
The author thanks Reviewer 1 for his reasonable suggestions and valuable comments on this article. Based on the issues raised by Reviewer 1, the authors have thoroughly and carefully checked and corrected the preface and full text of the manuscript. This submitted manuscript is listed to illustrate the demonstration revision.
Revised and responded to individually based on the valuable comments of the reviewers1. Here, in the introductory section and especially in the second half, the authors focus on the motivation for this paper and transparent thoughts on independent features. The authors have given the necessary rationale and incorporated the revisions in the corresponding sections of this manuscript (the revisions have been highlighted in blue).
Revised Supplement 1:
Numerous reported and researched results indicate that microtexturing technology does significantly improve the friction reduction properties and load-bearing capacity of the gear meshing contact interfaces. Multi-scale microtextures configurations can be designed on gear MTS to improve lubrication performance, and carefully selected microtextures are expected to help retain lubricating oil and enhance hydrodynamic effects at the gear meshing interface. Textured MTS are generated based on the TME model and "validated" by numerical simulation.
Revised Supplement 2:
This study describes the numerical simulation of multiscale microtextured MTS in TEHL contacts. The focus is on the selection and determination of MTS microtexturing distribution patterns based on TEHL performance. Contact multi-scale geometry, TME distribution density, interface materials and operating conditions all affect the performance of textured MTS. Evaluating the design rationality of an MTS microtextured requires extensive numerical simulations. A complete experimental evaluation of MTS microtextured properties is expensive and time-consuming. Multiscale MET techniques involving micro-textured MTS generation and analytical model-based performance evaluation are considered to be the most effective means to find generalized directions for TME creation and configuration optimization.
Revised Supplement 3:
In the process of high-speed and heavy-duty gear transmission systems (such as marine power rear transmission systems) operation, problems such as MTS lubrication performance-friction/wear behavior-carrying capacity will inevitably be encountered. At present, there are many technical solutions for solving the above problems. As an effective technical method, MTS microtextured technology has attracted extensive attention from scholars at home and abroad. In recent years, the contact interface micro-texture technology has achieved great results in the research of the meshing inter-face of gear pairs, which is of great significance for improving the lubrication performance-friction/wear behavior-load bearing capacity of MTS at high speed and heavy load.
Revised Supplement 4:
The results of years of research by scholars’ report that the lower the roughness of the meshing interface, the better the line contact performance. In contrast, the micro-textured MTS with multi-scale geometry exhibits better lubricating performance-friction/wear behavior-load bearing capacity.
Revised Supplement 5:
This topic analyzes the latest background and recent trends of the microtexture of the contact interface, emphatically pointing out the purpose and development status of this research. The mechanism of lubricating performance-friction/wear behavior- load bearing capacity of multi-scale micro-textured MTS is summarized, and the reason for the formation of the mechanism is explained. The TME configuration optimization of contact morphology and interface parameters is discussed, and the optimal microtexture parameter range of MTS is obtained.
Unrevised original reference ([27]):
- Arasan U., Marchetti F., Chevillotte F., Tanner G., Chronopoulos D., Gourdon E. On the accuracy limits of plate theories for vibro-acoustic predictions. Journal of Sound and Vibration 2021, Vol. 493, p. 115848.
Revised reference ([27]):
- Wang Y. J., Jacobs G., König F., Zhang S., Goeldel S. Investigation of Microflow Effects in Textures on Hydrodynamic Performance of Journal Bearings Using CFD Simulations. Lubricants 2023, Vol. 11, Issue 1, p. 20.
Q2. The description of the numerical model used for the simulation is also very hard to follow, extreme examples being the sentences
“In the simulation analysis of TEHL line contact, the approximation errors caused by considering the classical asymptotic assumptions are obvious and non- negligible, which, together with the differences in multi-scale properties, is difficult to accomplish, solving for more accurate numerical solutions from a complex set of equations.”
and
“Comparing this value of […] with […] and determine whether the initial assumption of a constant factor holds. If this does not hold, another constant factor is assumed and the determination continues until convergence is achieved […]”
Sincerely respond to Q2 raised by Reviewer 1:
The authors would like to thank Reviewer 1 for his sound advice and valuable comments on this manuscript. Based on the issues raised by Reviewer 1, the authors have thoroughly and carefully checked and revised this manuscript in accordance with Reviewer 1's suggestions. The authors have given the necessary explanations and elaborations in each case.
Case 1: “In the simulation analysis of TEHL line contact, the approximation errors caused by considering the classical asymptotic assumptions are obvious and non- negligible, which, together with the differences in multi-scale properties, is difficult to accomplish, solving for more accurate numerical solutions from a complex set of equations.”
What the authors are most interested in expressing is as follows.
In the simulation analysis of MTS line contact in the TEHL state, the approximation errors caused by considering the classical asymptotic assumptions are obvious and non-negligible, and together with the differences in the multiscale properties of the MTS microtexturing, the quest to obtain more accurate numerical solutions from a complex set of equations is clearly a difficult task.
Through this research project, the homogenised microhydrodynamics of the MTS with interfacial TME characteristics in the TEHL steady state is approximated accurately, the microgeometry of the multiscale structure with contact interface TME characteristics is simulated, the existence of an optimal geometry particularly effective for the load carrying capacity of the MTS is proposed and discussed, and the effectiveness of the optimized MTS with interfacial TME characteristics significantly influencing the load carrying capacity of the macroscopic characteristics of the generic slip linear contact is confirmed.
Case 2: “Comparing this value of […] with […] and determine whether the initial assumption of a constant factor holds. If this does not hold, another constant factor is assumed and the determination continues until convergence is achieved […]”
The authors have elaborated the necessary explanations as follows:
The Reynolds and elasticity equations are discretized by the finite element method using second order Lagrangian elements. The microscopic problem of the homogenised model is solved in a homogeneous manner at each node of the grid of the macroscopic problem. This leads to a large number of degrees of freedom, however, as our goal is to assess the accuracy of the model presented here, we wish to eliminate sources of error, for example, from decoupling the macroscopic scale equations from the microscopic equations. Nevertheless, this will limit our ability to solve small numerical problems. A Newton-Raphson solver is used for the non-linear system and all systems of linear equations are solved by the direct method. The results are calculated using commercial software.
In this work, we present a homogenised model for the MTS line contact problem in the steady-state TEHL state that considers piezoviscous effects and the variation of density with pressure, and where the size of the microtexturing MTS multiscale is assumed to be non-minimal. In cases of technical significance, microhydrodynamic effects are correctly captured. While these developments are for the one-dimensional stationary case in order to better assess the performance of the model, extension to the two-dimensional transient case is not difficult.
Q3. What constitutes the multi-scale character of the model, and what kind of homogenization techniques does it use? Eq. (4) is also very obscure.
Sincerely respond to Q3 raised by Reviewer 1:
The authors thank Reviewer 1 for sound suggestions and valuable comments on this manuscript. Based on the questions raised by Reviewer 1, the authors conducted a comprehensive and detailed inspection and correction of the full text as suggested by Reviewer 1, and explained the problems raised by reviewer 2 one by one.
In this sub-project, a gear friction dynamics model with the meshing Contact Interface Micro-Texture (CIMT) is proposed, which mainly includes gear dynamics models with different CIMT evolutions and gear 3D TEHL transient meshing models to explore the correlation effect of CIMT on MTS dynamic characteristics. The MTS friction dynamics coupling equations with CIMTs are solved by an iterative loop between the above two models, as shown in Auxiliary Figure 1. In the absence of iterative loop solution, first initialize the basic parameters of the MTS, such as the comprehensive error of gear tooth manufacturing, rated input torque and its corresponding speed. The lubricated MTS with Textured Micro-Elements (TME) is analyzed according to the potential energy method. If it is not iterated, the starting value for viscous damping and sliding/rolling friction are set to. Considering the mentioned above excitations, the vibration equation of the MTS dynamics model can be solved according to the Runge-Kutta analytical method. The sliding speed and curvature radius of the tooth surface at the meshing position are predicted by the MTS contact analysis.
Auxiliary Figure 1 Flowchart of CIMT calculation method
In this preliminary study, an infinitely long cylindrical contact is assumed. A section of the textured micro-element computational domain for solving the fluid dynamics coincides with the line, as shown in Auxiliary Figure 2, where, is the height of the texture micro element, is the aspect ratio of the micro element. In numerical simulations of lubricating contacts, the approximation errors caused by the classical asymptotic assumptions can be quite large, and the difference in scale leads to the solution of complex systems of equations. Assuming that the micro-scale is homogenous and periodic, based on the formal method of decoupling the macro-scale and micro-scale, a homogenized micro-elastohydrodynamic model is introduced, which considers the pressure and deformation that cannot be ignored at the micro-scale, and then the general applicability of the classical asymptotic homogenization method is extended.
Auxiliary Figure 2 A section of the textured micro-element computational domain
Equation (4) is solved in elastic deformation considering thermal effects. Note that the same formula can be applied even if the lower surface is rough, since we are considering equivalent solids.
The revised equation (4) is written as follows:
(4)
Q4. In the results section, due to the lack of experimental data or comparison with more principled numerical simulations, it is difficult to judge, whether the differences in tribological performance between the different analysed micro- textures are indeed larger than the margin of error of the numerical model. In fact, it is the task of the authors to prove (or, at least, substantiate) that these differences are significant, if their results shall be of any further use (the “finding” that micro- texture can be tribologically useful is not new at all).
Sincerely respond to Q4 raised by Reviewer 1:
The authors thank the reviewer 1 for reasonable suggestions and valuable comments on this manuscript. According to the questions raised by the reviewer 1, the author conducted a comprehensive and detailed inspection and correction of the full text according to the suggestions of the reviewer 1, and made necessary explanations to the questions raised by the reviewer 1 in order to expect the reviewer 1's approval and acceptance.
The surface pressure, film thickness distribution and temperature distribution of meshing teeth with interfacial microtexture are solved by multigrid integration method. The meshing interface pressure calculation, and thermo-elastic deformation term calculation caused by the pressure adopts the multi-grid integration method, and the interface temperature calculation adopts the column-by-column scanning method. The grid is divided into 5 layers, the highest grid is divided into 512 nodes in the x direction and 512 nodes in the y direction, the temperature gradient in the oil film between the interfaces is larger, and the equidistant grid is used, and the number of nodes is 10. In the solid, the temperature gradient is larger near the solid-liquid interface, and the temperature change tends to be gentle at the distance away from the solid-liquid interface. Therefore, unequal spacing grids are used, the grid spacing is a proportional sequence, and the number of nodes in both solids is 6. When the error of pressure before and after iteration is less than1×10-4, and the relative error of load and temperature is less than1×10-5, the iteration ends.
The distribution of the nonlinear strong coupling field (film thickness, film pressure, film temperature) at the microtexture interface is solved by the combination of the energy equation and the Reynolds equation. The conclusions in Auxiliary Figures 3 and 4 are as follows: The overall thickness of the oil film considering the thermal effect becomes thinner, especially the minimum film thickness, which decreases sharply after considering the thermal effect. After considering the thermal effect, the pressure fluctuation amplitude of the textured surface becomes larger, and the secondary pressure peak value increases significantly.
By optimizing the parameters related to the CIMT, the synergistic regulation of improved lubrication and increased load-bearing capacity. The proposed model is used to analyze the influence of the multi-scale structure and distribution parameters of the MTS microtexture on the TEHL characteristics, and the optimal distribution interval of the microtexture multi-scale structure parameters when the oil film properties of the MTS interface are in a good state under high-strength contact is obtained.
Auxiliary Figure 3 Interfacial film pressure curves of 3 textured micro-element configurations
Auxiliary Figure 4 Oil film thickness curves of interfacial films for 3 textured micro-element configurations
Q5. Last but not least, a reflecting discussion, acknowledging the possible drawbacks or improvements of the used methodology, is missing completely.
Sincerely respond to Q5 raised by Reviewer 1:
The authors are very grateful to Reviewer 1 for his sound suggestions and valuable comments on this article. Based on the issues raised by Reviewer 1, the authors have carried out a comprehensive and detailed inspection and revision of the Conclusion section according to Reviewer 1's suggestion.
Considering that the interfacial lubricating medium exhibits strong non-Newtonian effects due to instantaneous pressure and thermal effects in the MTS heavy-duty meshing state with TME structure characteristics, the optimized values of the multi-scale microtexture parameters are analyzed.
The next focus of future work is to conduct experimental research on the dynamic behavior of gear MTS with various contact interface micro-texture morphologies, so as to deepen the verification of related theoretical findings through the research of this sub-project.
Contribution of each individual co-author:
No conflict of interest exits in the submission of this manuscript, and manuscript is approved by all authors for publication. We would like to declare on behalf of our co-authors that the work described was original research that has not been published previously, and not under consideration for publication elsewhere, in whole or in part. All the authors listed have approved the manuscript that is enclosed.
In this subject research, we have proposed the “Numerical Analysis of Friction Reduction and ATSLB Capacity of Lubricated MTS with Textured Micro-Elements” for a theoretical model of thermoelastic lubricated interfacial Textured Micro-Element (TME) load-bearing contact, and the effective friction reduction and Anti-Thermoelastic Scuffing Load-Bearing (ATSLB) characteristics between random rough Meshing Teeth Surfaces (MTS) are investigated, the mechanism linking interfacial thermoelastic lubrication, TME meshing friction reduction and ATSLB is revealed. The MTS is pre-set as a line contact mode, which breaks through the limitation of assuming that the actual meshing area is much smaller than the nominal interface contact domain. The real contact domain area between MTS with multi-scale Micro-Element Textures (MET) is obtained for the numerical calculation of the three-dimensional equivalent TME contact volume, which is the correlation bridge between friction reduction and ATSLB of the thermoelastic lubrication interface.
Our main contribution to the field is to predict the time-varying behaviour of the textured meshing interface friction reduction with TME contact load under thermoelastic lubrication conditions by the proposed theoretical model. Numerical simulations show that the textured interface meshing volume is the key to solving the load-bearing problem of line contact between randomly rough teeth surfaces. The friction coefficients of the MTS are reduced by 13-24%. Interfacial MET parameters are optimized and the correlation between linear velocity, time-varying load and micro-configuration scale and meshing interface oil film thickness, oil film pressure distribution morphological trends and ATSLB capacity is elaborated. The lubricated load-bearing and friction reduction behaviour between the textured MTS is quantified by the thermoelastic voids of the TME interface and the actual meshing volume ratio, which provides a new perspective for further insight into the lubrication and friction reduction behaviour between the MTS with multi-scale MET-ATSLB coupling mechanism.
The novelty and significance of this manuscript are as follows:
The work was studied mainly “Numerical Analysis of Friction Reduction and ATSLB Capacity of Lubricated MTS with Textured Micro-Elements”. This topic involves the homogenized micro hydrodynamics of the MTS with interfacial TME characteristics in the TEHL steady state is approximated accurately, the microgeometry of the multiscale structure with contact interface TME features is simulated, the existence of an optimal geometry that is particularly effective for the load-bearing capacity of the MTS is proposed and discussed, and the validity of the optimized MTS with interfacial TME features significantly affecting the load-bearing capacity of the macroscopic properties of the generic slip linear contact is confirmed.
We are very hoped to publish this article in your journal, and I thank you on behalf of our group. We apologize for what we have not done well. We hope we will continue to submit better articles to you.
Thank you in advance for considering this revised submission. We very much look forward to your reply and any questions needed next.
Thank you and best regards.
Sincerely,
Corresponding author/First author
Xigui Wang Professor, PhD Supervisor
School of Engineering Technology, Northeast Forestry University, No. 26, Hexing Road, Xiangfang District, Harbin, 150040, China; School of Mechatronics and Automation, Huaqiao University, No. 668 Jimei Avenue, Jimei District, Fujian Province, Xiamen, 361021, China
Hui Huang
School of Mechatronics and Automation, Huaqiao University, No. 668 Jimei Avenue, Jimei District, Fujian Province, Xiamen, 361021, China;
Jingyu Song
School of Mechatronics and Automation, Huaqiao University, No. 668 Jimei Avenue, Jimei District, Fujian Province, Xiamen, 361021, China;
Yongmei Wang
School of Motorcar Engineering, Heilongjiang Institute of Technology, No. 999, Hongqidajie Road, Daowai District, Harbin, 150036, China; School of Mechatronics and Automation, Huaqiao University, No. 668 Jimei Avenue, Jimei District, Fujian Province, Xiamen, 361021, China
阮家福
东北林业大学工程技术学院, 哈尔滨市香坊区合兴路26号, 哈尔滨, 150040;华侨大学机电一体化与自动化学院, 福建省集美区集美大道668号, 厦门, 361021, 中国

Reviewer 2 Report
(1)Word “The” on page 2,line 78 should be deleted.
(2)On page 3, the curves given in figure 1 are not consistent with the title of the figure. The purpose to introduce figure 1 should be explained.
(3)Figure 4 on page 5 should be explained in a little detail.
(4)"mm" in line 289 on page 7 should be "μm" since the unit is about the width and depth of the micro-texture.
(5)On page 8, line 306 and 307, there is "The numerical calculation and simulation data of the experimental parameters are presented in Table 1", which parameters are about the numerical simulation, which are about the simulation data of the experimental in table 1. What is the difference between numerical simulation and simulation experiment in the manuscript.
(6)The temperature of the oil film should be given.
(7)"Depth =15.0 μm "in Fig. 9 should be "Depth =20 μm" since in line 345 there is"The behavior of three TME depths of 5.0, 10 and 20 μm corresponding to different interface TME aspect ratios of 0.1, 0.15 and 0.2 are analyzed and compared".
(7)“TMS” in line 377 should be "MTS".
(8)In line 387~390, there is "A further thought-provoking fact revealed in Figure 10 is that the scale variation of the maximum MTS load-bearing contact pressure with TME depth in the TEHL steady state does not fully depict that a meshing interface with a deeper scale TME necessarily leads o better MTS tribological properties." It is suggested that authors should explain in detail according to the Fig. 10.
(9)It is suggested that the author should compare the test results of the actual MTS contact test with and without TME to verify the conclusions of the manuscript.
Author Response
January 19, 2023
Dear editor-in-chief
The open access journal Lubricants
We have submitted a research article hoping to be published in the journal Lubricants, titled “Numerical Analysis of Friction Reduction and ATSLB Capacity of Lubricated MTS with Textured Micro-Elements.” The paper was coauthored by XiguiWang, Hui Huang, Jingyu Song, Yongmei Wang, and Jiafu Ruan. We have checked the manuscript and revised it according to the comments. Overall, the comments have been fair, encouraging and constructive. We have learned much from it. We submit here the revised manuscript to meet the evaluation conditions and requirements of the reviewers.
Response to Reviewers' Comments:
Reviewer2:
On behalf of all members of our team, I would like to thank the Reviewer 2 for approving and accepting this manuscript. The authors have once again fully reviewed and refined this revised manuscript, and reflected the refined content in this article. In the long-term in the future, our research team will submit more excellent and high-quality articles to your journal.
Q1. Word “The” on page 2, line 78 should be deleted.
Sincerely respond to Q1 raised by Reviewer 2:
The authors have made revisions as suggested by Reviewer 2 and reflected in this manuscript.
Q2. On page 3, the curves given in figure 1 are not consistent with the title of the figure. The purpose to introduce figure 1 should be explained.
Sincerely respond to Q2 raised by Reviewer 2:
The authors have followed reviewer 2's suggestion to explain the presentation and these contents have been reflected in this manuscript. The purpose of introducing Figure 1 is explained as follows:
This topic analyzes the latest background and recent trends of the microtexture of the contact interface, emphatically pointing out the purpose and development status of this research. The mechanism of lubricating performance-friction/wear behavior-load bearing capacity of multi-scale micro-textured MTS is summarized, and the reason for the formation of the mechanism is explained. The TME configuration optimization of contact morphology and interface parameters is discussed, and the optimal microtexture parameter range of MTS is obtained.
Q3. Figure 4 on page 5 should be explained in a little detail.
Sincerely respond to Q3 raised by Reviewer 2:
The authors have fully accepted reviewer 2's suggestion and have explained Figure 4 in more detail below.
In this sub-project, a gear friction dynamics model with the meshing Contact Interface Micro-Texture (CIMT) is proposed, which mainly includes gear dynamics models with different CIMT evolutions and gear 3D TEHL transient meshing models to explore the correlation effect of CIMT on MTS dynamic characteristics. The MTS friction dynamics coupling equations with CIMTs are solved by an iterative loop between the above two models, as shown in Auxiliary Figure 1. In the absence of iterative loop solution, first initialize the basic parameters of the MTS, such as the comprehensive error of gear tooth manufacturing, rated input torque and its corresponding speed. The lubricated MTS with Textured Micro-Elements (TME) is analyzed according to the potential energy method. If it is not iterated, the starting value for viscous damping and sliding/rolling friction are set to. Considering the mentioned above excitations, the vibration equation of the MTS dynamics model can be solved according to the Runge-Kutta analytical method. The sliding speed and curvature radius of the tooth surface at the meshing position are predicted by the MTS contact analysis.
Auxiliary Figure 1 Flowchart of CIMT calculation method
In this preliminary study, an infinitely long cylindrical contact is assumed. A section of the textured micro-element computational domain for solving the fluid dynamics coincides with the line, as shown in Auxiliary Figure 2, where, is the height of the texture micro element, is the aspect ratio of the micro element. In numerical simulations of lubricating contacts, the approximation errors caused by the classical asymptotic assumptions can be quite large, and the difference in scale leads to the solution of complex systems of equations. Assuming that the micro-scale is homogenous and periodic, based on the formal method of decoupling the macro-scale and micro-scale, a homogenized micro-elastohydrodynamic model is introduced, which considers the pressure and deformation that cannot be ignored at the micro-scale, and then the general applicability of the classical asymptotic homogenization method is extended.
Auxiliary Figure 2 A section of the textured micro-element computational domain
Q4. "mm" in line 289 on page 7 should be "μm" since the unit is about the width and depth of the micro-texture.
Sincerely respond to Q4 raised by Reviewer 2:
The authors have made revisions as suggested by Reviewer 2 and reflected in this manuscript.
Q5. On page 8, line 306 and 307, there is "The numerical calculation and simulation data of the experimental parameters are presented in Table 1", which parameters are about the numerical simulation, which are about the simulation data of the experimental in table 1. What is the difference between numerical simulation and simulation experiment in the manuscript.
Sincerely respond to Q5 raised by Reviewer 2:
The authors have followed reviewer 2's suggestion and have given the necessary explanations.
Simulation analysis numerically investigates the interfacial textured micro-element (TME) load bearing contact with thermoelastic lubrication, presents a theoretical model and proposes an effective friction reduction and resistance to thermoelastic scuffing load bearing capacity (ATSLB) between random rough mesh tooth surfaces (MTS), revealing the mechanism linking interfacial thermoelastic lubrication, TME mesh friction reduction and ATSLB. The actual contact domain area between MTS with multi-scale micro-element textures (MET) is used for the numerical calculation of the three-dimensional equivalent TME contact volume, which is the bridge between thermoelastically lubricated interfacial friction reduction and ATSLB. The proposed theoretical model predicts the time-varying behaviour of the textured meshing interface friction reduction with respect to the TME contact load under thermoelastic lubrication conditions. Numerical simulations show that the amount of textured interface engagement is the key to solving the load-bearing problem of line contact between randomly rough tooth surfaces. the friction coefficient of the MTS is reduced by 13-24%. The quantification of the lubrication load-bearing and friction reduction behaviour between textured MTS by means of the thermoelastic void and actual engagement volume ratio at the TME interface provides a new perspective for further understanding of the lubrication and friction reduction behaviour between MTS with a multi-scale MET-ATSLB coupling mechanism.
The numerical accuracy of the multiphysics, multiplatform model is evaluated through a mesh-independent study of the minimum film thickness for thermal steady state. TEHL simulations were performed on meshes with radial element counts of 34, 85, 170, 200, 250, and 340, with axial elements appropriately scaled. Since both the chamber model for deformation and the thermomechanical FEA model assume axisymmetric, the number of tangential elements and axial fluid elements are kept at 30 and 20, respectively. The model incorporates the effects of mechanical deformation, thermoelastic deformation through FEA-based influence coefficients, fluid flow modeled through the Reynolds equation, and heat transport modeled through the thin film energy equation.
Numerical simulation is the validation of numerical calculations based on analytical results of commercial software. The simulation experiment is based on the evaluation and verification of lubrication performance-friction/wear effect-carrying capacity on the type test bench proposed in this paper, which is a check of the numerical simulation of the scaled-down actual working conditions.
Q6. The temperature of the oil film should be given.
Sincerely respond to Q6 raised by Reviewer 2:
The authors have followed reviewer 2's suggestion and have given the necessary explanations.
The surface pressure, film thickness distribution and temperature distribution of meshing teeth with interfacial microtexture are solved by multigrid integration method. The meshing interface pressure calculation, and thermo-elastic deformation term calculation caused by the pressure adopts the multi-grid integration method, and the interface temperature calculation adopts the column-by-column scanning method. The grid is divided into 5 layers, the highest grid is divided into 512 nodes in the x direction and 512 nodes in the y direction, the temperature gradient in the oil film between the interfaces is larger, and the equidistant grid is used, and the number of nodes is 10. In the solid, the temperature gradient is larger near the solid-liquid interface, and the temperature change tends to be gentle at the distance away from the solid-liquid interface. Therefore, unequal spacing grids are used, the grid spacing is a proportional sequence, and the number of nodes in both solids is 6. When the error of pressure before and after iteration is less than1×10-4, and the relative error of load and temperature is less than1×10-5, the iteration ends.
The distribution of the nonlinear strong coupling field (film thickness, film pressure, film temperature) at the microtexture interface is solved by the combination of the energy equation and the Reynolds equation. The conclusions in Auxiliary Figures 3 and 4 are as follows: The overall thickness of the oil film considering the thermal effect becomes thinner, especially the minimum film thickness, which decreases sharply after considering the thermal effect. After considering the thermal effect, the pressure fluctuation amplitude of the textured surface becomes larger, and the secondary pressure peak value increases significantly.
By optimizing the parameters related to the CIMT, the synergistic regulation of improved lubrication and increased load-bearing capacity. The proposed model is used to analyze the influence of the multi-scale structure and distribution parameters of the MTS microtexture on the TEHL characteristics, and the optimal distribution interval of the microtexture multi-scale structure parameters when the oil film properties of the MTS interface are in a good state under high-strength contact is obtained.
Auxiliary Figure 3 Interfacial film pressure curves of 3 textured micro-element configurations
Auxiliary Figure 4 Oil film thickness curves of interfacial films for 3 textured micro-element configurations
Q7. "Depth =15.0 μm "in Fig. 9 should be "Depth =20 μm" since in line 345 there is” The behavior of three TME depths of 5.0, 10 and 20 μm corresponding to different interface TME aspect ratios of 0.1, 0.15 and 0.2 are analyzed and compared".
Sincerely respond to Q7 raised by Reviewer 2:
The authors have made revisions as suggested by Reviewer 2 and reflected in this manuscript.
Revised Figure 9:
Revised Figure 9 Variation curves of minimum TEHL oil film thickness for different TME depth scale sizes with different interface TME aspect ratios at a load of 200N and a slip linear velocity of 2.0m/s
Q8. “TMS” in line 377 should be "MTS".
Sincerely respond to Q8 raised by Reviewer 2:
The authors have made revisions as suggested by Reviewer 2 and reflected in this manuscript.
Q9. In line 387~390, there is "A further thought-provoking fact revealed in Figure 10 is that the scale variation of the maximum MTS load-bearing contact pressure with TME depth in the TEHL steady state does not fully depict that a meshing interface with a deeper scale TME necessarily leads o better MTS tribological properties." It is suggested that authors should explain in detail according to the Fig. 10.
Sincerely respond to Q9 raised by Reviewer 2:
The authors have followed reviewer 2's suggestion and have given the necessary explanations.
Numerous reported and studied results show that microfabrication techniques do significantly improve the friction reduction and load carrying capacity of the interface of contact parts. The multiscale geometrical parameters of microtexturing are one of the central factors in optimizing the benefits of microtexturing interfaces. It is emphasized that these known results provide significant improvements in terms of brief frictional wear and lubricant load-bearing. In most cases, the contact interface microfabrication is presented in a pre-engineered form. Undoubtedly, each complexity parameter correlation and mechanistic understanding is not yet fully understood, and the analytical descriptions are only in a limited number of hypothetical cases, failing to break through most of the known mechanisms limited to discrete textures of crater area ratios and pattern shapes, and furthermore, little pre-research work has been done on multi-scale contact interface textures from the perspective of geometric parameter characterization.
Assuming that multiscale texturing interfaces are microhomogeneous and periodic, a formal approach based on the decoupling of macroscale and microscale is introduced to introduce a micro-element homogenised interface micro TEHL dynamic pressure contact model that considers the non-negligible deformation due to compressive load bearing and thermal effects at the microscale, which extends the general applicability of the classical progressive homogenization approach. The correlative effect of MTS texturization on the tribological performance of TEHL slip linear contact is investigated by means of simulated experiments and a numerical computational model which analyses and predicts the load carrying capacity of MTS with and without interfacial TME features, which is used to assess the friction coefficients of micro-textured MTS.
The search for correlation between key optimal micro-element shape design parameters and geometric scale parameters is necessary to obtain maximum frictional wear reduction benefits and thus improve performance and pursue this concept for lubricated load-bearing engineering applications at gear meshing interfaces. In this study, pre-designed micro-element texturing patterns of slip line contact interfaces under thermoelastic flow lubrication conditions reveal the correlation between friction and load carrying capacity of key parameters such as depth and width under mixed thermoelastic hydrodynamic lubrication.
This apparent improvement is more pronounced for MTS at lower applied loads and higher linear velocities of slip. These interfacial TME features, if the MTS has an appropriate micro-texture multiscale geometry, which can significantly improve the thermoelastic lubrication behaviour and load-bearing capacity, reducing the coefficient of friction by 13-24% compared to normal MTS (no interfacial TME features). The slip linear velocity plays an important role in the formation of TEHL oil film thickness and MTS load-bearing capacity. At slip contacts with low linear velocities, the presence of interfacial TME features leads to a reduction in TEHL oil film thickness, which weakens the MTS load-bearing capacity. As the linear velocity of interfacial slip increases, the TME multiscale features present a rather interesting representation of TEHL oil film thickness and MTS load-bearing capacity.
Q10. It is suggested that the author should compare the test results of the actual MTS contact test with and without TME to verify the conclusions of the manuscript.
Sincerely respond to Q10 raised by Reviewer 2:
The authors have followed reviewer 2's suggestion and have given the necessary explanations.
Finding the correlation between the key optimum micro-element shape design parameters and geometric scale parameters is necessary to obtain the maximum frictional wear reduction benefit and thus improve performance, and to pursue this concept for lubrication load bearing engineering applications at gear meshing interfaces. In this study, micro-element texturing patterns of pre-designed slip line contact interfaces under thermoelastic flow lubrication conditions reveal the correlation between friction and load carrying capacity of key parameters such as depth and width under mixed thermoelastic hydrodynamic lubrication. Assuming that the multi-scale woven interface is microhomogeneous and periodic, a formal approach based on macro-scale and micro-scale decoupling is used to introduce a micro-TEHL dynamic-pressure contact model for the micro-element homogenised interface, which extends the general applicability of the classical progressive homogenization approach by taking into account the non-negligible deformation due to compressive load-bearing and thermal effects at the micro-scale. The relevant effects of MTS texture on the tribological performance of TEHL slip-linear contacts are investigated through simulated experiments and numerical computational modelling, which analyses and predicts the load carrying capacity of MTS with and without interfacial TME characteristics, for assessing the friction coefficient of microtextured MTS.
This is the key to the continued research of our group, and we will be verifying the real time working conditions of an engineered gear train with micro-textured meshing teeth surfaces through a real experimental platform in the near future.
Contribution of each individual co-author:
No conflict of interest exits in the submission of this manuscript, and manuscript is approved by all authors for publication. We would like to declare on behalf of our co-authors that the work described was original research that has not been published previously, and not under consideration for publication elsewhere, in whole or in part. All the authors listed have approved the manuscript that is enclosed.
In this subject research, we have proposed the “Numerical Analysis of Friction Reduction and ATSLB Capacity of Lubricated MTS with Textured Micro-Elements” for a theoretical model of thermoelastic lubricated interfacial Textured Micro-Element (TME) load-bearing contact, and the effective friction reduction and Anti-Thermoelastic Scuffing Load-Bearing (ATSLB) characteristics between random rough Meshing Teeth Surfaces (MTS) are investigated, the mechanism linking interfacial thermoelastic lubrication, TME meshing friction reduction and ATSLB is revealed. The MTS is pre-set as a line contact mode, which breaks through the limitation of assuming that the actual meshing area is much smaller than the nominal interface contact domain. The real contact domain area between MTS with multi-scale Micro-Element Textures (MET) is obtained for the numerical calculation of the three-dimensional equivalent TME contact volume, which is the correlation bridge between friction reduction and ATSLB of the thermoelastic lubrication interface.
Our main contribution to the field is to predict the time-varying behaviour of the textured meshing interface friction reduction with TME contact load under thermoelastic lubrication conditions by the proposed theoretical model. Numerical simulations show that the textured interface meshing volume is the key to solving the load-bearing problem of line contact between randomly rough teeth surfaces. The friction coefficients of the MTS are reduced by 13-24%. Interfacial MET parameters are optimized and the correlation between linear velocity, time-varying load and micro-configuration scale and meshing interface oil film thickness, oil film pressure distribution morphological trends and ATSLB capacity is elaborated. The lubricated load-bearing and friction reduction behaviour between the textured MTS is quantified by the thermoelastic voids of the TME interface and the actual meshing volume ratio, which provides a new perspective for further insight into the lubrication and friction reduction behaviour between the MTS with multi-scale MET-ATSLB coupling mechanism.
The novelty and significance of this manuscript are as follows:
The work was studied mainly “Numerical Analysis of Friction Reduction and ATSLB Capacity of Lubricated MTS with Textured Micro-Elements”. This topic involves the homogenized micro hydrodynamics of the MTS with interfacial TME characteristics in the TEHL steady state is approximated accurately, the microgeometry of the multiscale structure with contact interface TME features is simulated, the existence of an optimal geometry that is particularly effective for the load-bearing capacity of the MTS is proposed and discussed, and the validity of the optimized MTS with interfacial TME features significantly affecting the load-bearing capacity of the macroscopic properties of the generic slip linear contact is confirmed.
We are very hoped to publish this article in your journal, and I thank you on behalf of our group. We apologize for what we have not done well. We hope we will continue to submit better articles to you.
Thank you in advance for considering this revised submission. We very much look forward to your reply and any questions needed next.
Thank you and best regards.
Sincerely,
Corresponding author/First author
Xigui Wang Professor, PhD Supervisor
School of Engineering Technology, Northeast Forestry University, No. 26, Hexing Road, Xiangfang District, Harbin, 150040, China; School of Mechatronics and Automation, Huaqiao University, No. 668 Jimei Avenue, Jimei District, Fujian Province, Xiamen, 361021, China
Hui Huang
School of Mechatronics and Automation, Huaqiao University, No. 668 Jimei Avenue, Jimei District, Fujian Province, Xiamen, 361021, China;
Jingyu Song
School of Mechatronics and Automation, Huaqiao University, No. 668 Jimei Avenue, Jimei District, Fujian Province, Xiamen, 361021, China;
Yongmei Wang
School of Motorcar Engineering, Heilongjiang Institute of Technology, No. 999, Hongqidajie Road, Daowai District, Harbin, 150036, China; School of Mechatronics and Automation, Huaqiao University, No. 668 Jimei Avenue, Jimei District, Fujian Province, Xiamen, 361021, China
Jiafu Ruan
School of Engineering Technology, Northeast Forestry University, No. 26, Hexing Road, Xiangfang District, Harbin, 150040, China; School of Mechatronics and Automation, Huaqiao University, No. 668 Jimei Avenue, Jimei District, Fujian Province, Xiamen, 361021, China

Round 2
Reviewer 1 Report
The manuscript has not been properly revised. While the issues raised in the previous review were (to some extent) addressed in the "answers to reviewers", most of that doesn't reflect in the manuscript. In fact, except for the introductory section, almost nothing was changed in the manuscript for the revision.
Also, sections marked in blue (for changes to the manuscript), are really the same as in the previous version, which I consider highly unprofessional.
In view of that, I refuse to review the manuscript, unless a proper revision is submitted.
Author Response
Author's Reply to the Review Report (Reviewer 2) (2023-01-25)
Q1 The manuscript has not been properly revised. While the issues raised in the previous review were (to some extent) addressed in the "answers to reviewers", most of that doesn't reflect in the manuscript. In fact, except for the introductory section, almost nothing was changed in the manuscript for the revision.
Sincerely respond to Q1 raised by Reviewer 2:
The authors are very grateful to reviewer 2 for their valuable suggestions and gladly accepted them. The authors carefully revised the responses to the reviewers considering the issues raised in the previous review and reflected the revised contents in this forthcoming manuscript. The authors elaborated based on the relevant facts previously studied by the research group to support the research basis and analysis data of this manuscript. The authors have listed the revised content in this manuscript.
Revised content 1:
The research purpose of this subject is on the of multi-scale microtextures distribution patterns is engineered into MTS for lubrication performance improvement and friction reduction and ATSLB capacity.
Revised content 2:
the multiple scales are contacted by wear (which is stochastic in this case) or by micro-deformation due to thermoelastic lubrication in a wear-free process.
Revised content 3:
Fully designed and optimized MTS textured configurations are considered as a prom-ising approach to improve the ATSLB capacity and reduce slip-line contact friction losses. The purpose of this topic is to analyze the local lubrication mechanism of MTS thermoelastic interfacial TME load-bearing contact from a microfluidic perspective, while recognizing the relationship between multi-scale micro-element textures and oil film formation in the entire MTS interaction. For this research aim, a hydrodynamic lubrication model of micro-textured MTS is engineered.
Q2 Also, sections marked in blue (for changes to the manuscript), are really the same as in the previous version, which I consider highly unprofessional.
Sincerely respond to Q2 raised by Reviewer 2:
The authors are very grateful for the sound advice and valuable comments from Reviewer 2. The authors have marked the revised content section of this manuscript in blue to enhance the professionalism of the subject group in future manuscript submissions and revisions.
The authors have revised the manuscript again using the version of the manuscript in the link mentioned.
Q(I) Please check that all references are relevant to the contents of the manuscript.
Sincerely respond to Q (I) raised by Reviewer:
The authors have carefully checked all references and confirmed their relevance to the content of the manuscript.
Q(II) Any revisions to the manuscript should be marked up using the “Track Changes” function if you are using MS Word/LaTeX, such that any changes can be easily viewed by the editors and reviewers.
Sincerely respond to Q (II) raised by Reviewer:
The authors have marked any changes to the manuscript using the 'track changes' feature (revisions to this manuscript have been marked in blue) so that editors and reviewers can easily view any changes.
Q(III) Please provide a cover letter to explain, point by point, the details of the revisions to the manuscript and your responses to the referees’ comments.
Sincerely respond to Q (III) raised by Reviewer:
The authors have provided a cover letter and have provided point-by-point details of the changes made to the manuscript and responses to the referee's comments.
Q(IV) If you found it impossible to address certain comments in the review reports, please include an explanation in your appeal.
Sincerely respond to Q (IV) raised by Reviewer:
The authors have provided the necessary and reasonable explanations for the comments made in the review of this manuscript.
Q(V) The revised version will be sent to the editors and reviewers.
Sincerely respond to Q (V) raised by Reviewer:
The authors have sent the revised version to the editors and reviewers.
Contribution of each individual co-author:
No conflict of interest exits in the submission of this manuscript, and manuscript is approved by all authors for publication. We would like to declare on behalf of our co-authors that the work described was original research that has not been published previously, and not under consideration for publication elsewhere, in whole or in part. All the authors listed have approved the manuscript that is enclosed.
In this subject research, we have proposed the “Numerical Analysis of Friction Reduction and ATSLB Capacity of Lubricated MTS with Textured Micro-Elements” for a theoretical model of thermoelastic lubricated interfacial Textured Micro-Element (TME) load-bearing contact, and the effective friction reduction and Anti-Thermoelastic Scuffing Load-Bearing (ATSLB) characteristics between random rough Meshing Teeth Surfaces (MTS) are investigated, the mechanism linking interfacial thermoelastic lubrication, TME meshing friction reduction and ATSLB is revealed. The MTS is pre-set as a line contact mode, which breaks through the limitation of assuming that the actual meshing area is much smaller than the nominal interface contact domain. The real contact domain area between MTS with multi-scale Micro-Element Textures (MET) is obtained for the numerical calculation of the three-dimensional equivalent TME contact volume, which is the correlation bridge between friction reduction and ATSLB of the thermoelastic lubrication interface.
Our main contribution to the field is to predict the time-varying behaviour of the textured meshing interface friction reduction with TME contact load under thermoelastic lubrication conditions by the proposed theoretical model. Numerical simulations show that the textured interface meshing volume is the key to solving the load-bearing problem of line contact between randomly rough teeth surfaces. The friction coefficients of the MTS are reduced by 13-24%. Interfacial MET parameters are optimized and the correlation between linear velocity, time-varying load and micro-configuration scale and meshing interface oil film thickness, oil film pressure distribution morphological trends and ATSLB capacity is elaborated. The lubricated load-bearing and friction reduction behaviour between the textured MTS is quantified by the thermoelastic voids of the TME interface and the actual meshing volume ratio, which provides a new perspective for further insight into the lubrication and friction reduction behaviour between the MTS with multi-scale MET-ATSLB coupling mechanism.
The novelty and significance of this manuscript are as follows:
The work was studied mainly “Numerical Analysis of Friction Reduction and ATSLB Capacity of Lubricated MTS with Textured Micro-Elements”. This topic involves the homogenized micro hydrodynamics of the MTS with interfacial TME characteristics in the TEHL steady state is approximated accurately, the microgeometry of the multiscale structure with contact interface TME features is simulated, the existence of an optimal geometry that is particularly effective for the load-bearing capacity of the MTS is proposed and discussed, and the validity of the optimized MTS with interfacial TME features significantly affecting the load-bearing capacity of the macroscopic properties of the generic slip linear contact is confirmed.
We are very hoped to publish this article in your journal, and I thank you on behalf of our group. We apologize for what we have not done well. We hope we will continue to submit better articles to you.
Thank you in advance for considering this revised submission. We very much look forward to your reply and any questions needed next.
Thank you and best regards.
Sincerely,
Corresponding author/First author
Xigui Wang Professor, PhD Supervisor
School of Engineering Technology, Northeast Forestry University, No. 26, Hexing Road, Xiangfang District, Harbin, 150040, China; School of Mechatronics and Automation, Huaqiao University, No. 668 Jimei Avenue, Jimei District, Fujian Province, Xiamen, 361021, China
Hui Huang
School of Mechatronics and Automation, Huaqiao University, No. 668 Jimei Avenue, Jimei District, Fujian Province, Xiamen, 361021, China;
Jingyu Song
School of Mechatronics and Automation, Huaqiao University, No. 668 Jimei Avenue, Jimei District, Fujian Province, Xiamen, 361021, China;
Yongmei Wang
School of Motorcar Engineering, Heilongjiang Institute of Technology, No. 999, Hongqidajie Road, Daowai District, Harbin, 150036, China; School of Mechatronics and Automation, Huaqiao University, No. 668 Jimei Avenue, Jimei District, Fujian Province, Xiamen, 361021, China
Jiafu Ruan
School of Engineering Technology, Northeast Forestry University, No. 26, Hexing Road, Xiangfang District, Harbin, 150040, China; School of Mechatronics and Automation, Huaqiao University, No. 668 Jimei Avenue, Jimei District, Fujian Province, Xiamen, 361021, China

Round 3
Reviewer 1 Report
Please see the attached review file.

Author Response
Review for the manuscript lubricants-2163701-v3 entitled “Numerical Analysis of Friction Reduction and ATSLB Capacity of Lubricated MTS with Textured Micro-Elements”
Response to review comments for manuscript lubricant-2163701-v3:
January 28, 2023
Dear editor-in-chief
The open access journal Lubricants
We have submitted a research article hoping to be published in the journal Lubricants, titled “Numerical Analysis of Friction Reduction and ATSLB Capacity of Lubricated MTS with Textured Micro-Elements.” The paper was coauthored by XiguiWang, Hui Huang, Jingyu Song, Yongmei Wang, and Jiafu Ruan. We have checked the manuscript and revised it according to the comments. Overall, the comments have been fair, encouraging and constructive. We have learned much from it. We submit here the revised manuscript to meet the evaluation conditions and requirements of the reviewers.
Response to Reviewers' Comments:
Q1. ll. 49ff: revise “is on the of multi-scale […] ATSLB capacity.”
The research purpose of this subject is to engineer multi-scale microtexture distribution patterns into MTS to improve lubricating performance, reduce friction and increase ATSLB capacity.
Q2. ll. 55f: revise “formation of the contact interface voids formation”
which is just one result of the formation of voids at the contact interface.
Q3. l. 69: revise “contact interface microtextured” (at several instances in the manuscript)
The authors have revised "contact interface microtextured" to "microtextured contact interface". A total of three have been made in this manuscript.
Q4. ll. 69f: “Early similar reports date back to 1966” cite original source
Early similar reports date back to 1966 [4], the authors have revised according to the reviewer's comments and incorporated reference 4.
Q5. ll. 75f: “which inspires pioneering […] which is just an unveiling.”; revise for better readability
which inspires pioneering research of contact tribology in microtextured contact at the laser interface, while also is just an unveiling.
Q6. l. 95: revise “MTS microtextured” (at several instances in the manuscript)
The authors have revised "MTS microtextured" to "microtextured MTS". A total of three have been made in this manuscript.
Q7. L. 111: revise “a lubricated column pin-on-disc experiments are conducted”
a lubricated column pin-on-disc experiments is presented.
Q8. Ll. 112ff: revise complete passage “These results show that […] revealed by the Stribeck curves”
These results suggest that the multi-scale distribution patterns of microtextured, such as dimple size and dimple density, has a profound effect on the contact tribological properties of interface morphology compared to dimple geometry. According to the above report, the authors speculate that the multi-scale depth micro-dimples with micro-texture interface are more effective in increasing the thickness of the lubricating oil film, which is expected to improve the load-bearing capacity of the interfacial oil film in TEHL contact. Based on pin-disk experiments with multi-scale microtexture geometries (a series of square lattice micropores and parallel TME groove samples), the experimental results illustrated by the Stribeck curves
Q9. L. 141f: revise “distribution morphological density” (at several instances in the manuscript)
The authors have revised “distribution morphological density” to “morphological density distribution”. A total of two have been made in this manuscript.
Q10. L. 149: revise “multiscale indentations microtextured”
The authors have revised “multiscale indentations microtextured” to “multiscale indentations microtextures”.
Q11. ll. 157f: revise “better lubricating performance-friction/wear behavior-load bearing capacity”
the expected lubrication performance-friction/wear behavior-load bearing capacity.
Q12. ll. 163f: revise “local slip length variations in slip length caused”
local slip length variations in MTS line contact caused
Q13. ll. 172 f: revise “The mechanism of […] multi-scale micro-textured MTS”
Lubrication performance-friction/wear behavior-load bearing capacity mechanism of multi-scale micro-textured MTS
Q14. l.179: revise “Interface microtextured” (at several instances in the manuscript)
The authors have revised "Interface microtextured" to "Microtextured interface". A total of three have been made in this manuscript.
Q15. ll. 180 ff: “Numerous reported […] [30]”; this is an unnecessary repetition of the introduction
Microtexturing technology does significantly improve the friction reduction properties and load-bearing capacity of the contact interfaces [30].
Q16. ll. 185 ff: revise complete passage “In most cases […] geometric parameter characterization.”
The microtextured contact interface is presented in a pre-designed engineering. Need-less to say, the correlation and mechanistic understanding of each complexity parameter has not been fully explored, the analytical descriptions are only in a confined number of hypothetical cases and failing to break through most of the known mechanisms, being limited to crater area ratios and patterned shapes of discrete microelements textures, furthermore, there is little pre-research work on multi-scale contact interface microtextures from the perspective of geometric parameter characterization.
Q17. l. 193: revise “Contact interface microtextured types are generally divided into two types”
Microtextured contact interface forms are generally divided into two types, namely,
Q18. ll. 196 f.: “The model explains why, […] parallel straight lines”; this is unclear, revise
The continuous microtextures in this model are in the form of parallel straight lines or arrays of crossed curves.
Q19. ll. 210 f: revise “As micro-geometry becomes a deeply […] behaviour,”
As microscopic geometry emerges as insights into mechanisms of anomalous behavior,
Q20. ll. 220 f: revise “The effective cross sectional […] meshing interface”
The TEHL region of the effective cross-section of the column pin and the disc is simulated as a linear contact of the gear pair meshing interface,
Q21. Eqs. (1) and (2): ρm and H undefined, U unused
(1)
where the parameters are described as:
(2)
Here, a denotes half-width of the MTS contact region, represents the maximum Hertzian pressure P, is the central oil film thicknesses H, is lubricating oil density at Hertzian pressure P, is the friction coefficient mean value, is the viscosity of lubricating oil, is the average viscosity of lubricating oil, is the equivalent elastic modulus, U represents the scale factor.
Q22. l. 227: “a” must be italic
The authors have determined that "a" is in italics.
Q23. ll. 237 ff: revise complete passage “In the simulation analysis […] set of equations”
In the simulation analysis of TEHL line contact, the approximation errors caused by considering the classical asymptotic assumptions are obvious and non-negligible. Coupled with the differences in multiscale properties, it is difficult to achieve, to find more accurate numerical solutions from a complex set of equations.
Q24. ll. 244 ff: “Assuming that multi-scale textured […] progressive homogenizations methods”; this is unclear; elaborate about the decoupling of scales and homogenization
This issue has been addressed in previous papers published by the subject group and the authors have given the necessary explanatory statements as follows.
In this project, a gear friction dynamics model with Contact Interface Micro-Texture (CIMT) is proposed, which mainly includes gear dynamics models with different CIMT evolutions and gear 3D TEHL transient meshing models to explore the correlation effect of CIMT on Gear Transmission System (GTS) dynamic characteristics. The GTS friction dynamics coupling equations with CIMTs are solved by an iterative loop between the above two models, as shown in Auxiliary Figure 1. In the absence of iterative loop solution, first initialize the basic parameters of the Meshed Gear Pair (MGP), such as the comprehensive error of gear tooth manufacturing, rated input torque and its corresponding speed. The Time-Varying Meshing Stiffness (TVMS) is analyzed according to the potential energy method. If it is not iterated, the starting value for viscous damping and sliding/rolling friction are set to. Considering the mentioned above excitations, the vibration equation of the MGP dynamics model can be solved according to the Runge-Kutta analytical method. The sliding speed and curvature radius of the tooth surface at the meshing position are predicted by the GTS vibration motion analysis.
Auxiliary Fig. 1 Flowchart of CIMT calculation method
Auxiliary Fig. 2 A section of the textured micro-element computational domain
In this preliminary study, an infinitely long cylindrical contact is assumed. A section of the textured micro-element computational domain for solving the fluid dynamics coincides with the line, as shown in Auxiliary Figure 2, where, is the height of the texture micro element, is the aspect ratio of the micro element. In numerical simulations of lubricating contacts, the approximation errors caused by the classical asymptotic assumptions can be quite large, and the difference in scale leads to the solution of complex systems of equations. Assuming that the micro-scale is homogenous and periodic, based on the formal method of decoupling the macro-scale and micro-scale, a homogenized micro-elastohydrodynamic model is introduced, which considers the pressure and deformation that cannot be ignored at the micro-scale, and then the general applicability of the classical asymptotic homogenization method is extended.
At the same time, the Dynamic Meshing Force (DMF) is deduced by using the vibration response of the GTS, and then substituted into the preset load model (distributed form) to derive the DMF of the MGP. For the instantaneous values of,,, as well as the CIMT and lubricant properties, a typical semi-system method is used to analyze the control equations of the hybrid TEHL model. The rolling friction, the sliding friction and the viscous damping of the gear meshing interface are subject to the convergent solution of the hybrid TEHL model, which is used to calculate the updated dynamic response of the MGP dynamic model proposed above.
An iterative cycle process is always executed until the DMF of the MGP must meet the convergence criterion. The criterion for convergence of the iterative loop is as follows:
Auxiliary (1)
whererepresents the tooth pair mesh position in a meshing period, k the number of steps in the loop iteration, denotes the calculated DMF at (instantaneous meshing time) calculated in the K-th loop iteration step, and the parameter err is a pre-defined convergence threshold, whose predetermined value is set to in the research issue. So, it can be deduced, whenever the relative error value between the DMFs analyzed in two consecutive steps is not greater than the pre-defined convergence threshold, once the iterative loop process stops, it is assumed that a stable solution is determined.
Auxiliary Fig. 3 Three cases of CIMT configuration of MTS. (a) Case 1, (b) Case 2, (c) Case 3
In the actual operation of GTS, the micro-appearance morphologies of MTS directly depend on the processing accuracy, manufacturing process, operation law and constituent materials, etc. The related discuss focuses on the three different MTS micro-appearance morphologies that exist in actual processing and manufacturing. It is set in advance that these three cases all have the same micro-asperity peak Rq (root mean square (RMS) roughness value of MTS) and the same wavelength L, however, the cross-sections of the two are not equal, and are marked by random permutation (Case 1), sinusoidal distribution (Case 2), and semi-ellipse designed configuration (Case 3), respectively, as outlined in Auxiliary Figure 3. The associated influence of CIMT on the friction characteristics and dynamic responses of the MTS is further revealed.
Auxiliary Figure 4 depicts a general spur gear dynamic model considering MTS friction. Herein, the representative parameters and, which describe separately the friction behavior of driving gear (pinion) and driven gear (bull gear). The transient meshing description of a spur meshing pair is modeled simultaneously by TVMS and viscous damping . The support bearing of each gear is set to real-time simulation of equivalent support stiffness and damping indicated by the x and y directions, the relevant parameters are , , , and , , , . The driving/driven gears in the dynamic models are represented by the equivalent simplified representation of a rigid body whose mass, moment of inertia and radius are equal to the gear base circle radius. Assume that the rotational and translational movements of the two gears in the y direction are coupled along the Line Of Action (LOA) by the spring damping unit. represents the TVMS of the MGP, anddescribes the equivalent meshing damping of the MGP. is regarded as the Static Transmission Error (STE), which mainly includes gear tooth elastic deformation and gear tooth manufacturing error under static load conditions. The rotational and translational Degree of Freedoms (DoFs) of the MGP in the x-direction tends to be coupled in real time in the Off-Line Action (OLOA) direction. The control equations descriptions of the above gear dynamic model are expressed as:
Auxiliary (2)
Auxiliary (3)
Auxiliary (4)
Auxiliary (5)
Auxiliary (6)
Auxiliary (7)
whererepresents the number of the MGPs, denotes the external loads acting on the MGPs (, is shown as the pinon and bull gear respectively), and represents the torsional vibration velocity and acceleration of the MGPs , and denote the translational vibration velocity and acceleration in the OLOA direction of the MGPs , respectively, and represent the translational vibration velocity and acceleration in the LOA direction of the MGPs , respectively, represents the radius scalar at the meshing position of the MGPs . The frictional forces and of the driving/driven gears are described the hybrid TEHL calculation model. The DMF is expressed as:
Auxiliary (8)
Auxiliary Fig. 4A general spur gear dynamic model considering MTS friction behavior
The tooth pair number in the meshing region presents a periodic time-varying law during the meshing transient process of the spur MGP. Consider a MGP with a normal contact ratio between 1 and 2, in which the number of meshing gear teeth alternates. The entire meshing region is set as a Single Tooth Contact (STC) region and a Double Teeth Contact (DTC) region. The overall gear DMF is expressed as the sum of the DMF of all MGPs in the meshing region:
Auxiliary (9)
whereandrespectively denote the DMF of the first and second meshing gears teeth pairs, and represents the DMF when only one tooth pair in meshing. According to the load distribution model between meshing gear teeth, the DMF of each MGP is derived:
Auxiliary (10)
where is simulated by the TVMS of the MGP ().represents the viscous damping of the MGP (), which is usually assumed to be an associated parameter dependent on. The calculation formula can be expressed as:
Auxiliary (11)
where represents the initial vibration velocity of the meshing gear teeth relative to the parameter. In the above equation (11), and respectively denote the relative displacement and velocity of the MGP () along the LOA direction, which can be expressed as follows:
Auxiliary (12)
where indicates the displacement excitation caused by the tooth profile deviations (relative to the ideal involute tooth profile) for the MGP () during a MGP meshing transient process, whereas is shown as the derivative of the parameter , which is a quantitative characterization of the velocity excitation.
Q25. Eq. (4): no source or derivation is given; where are thermoelastic effects? what are the boundaries of the integral?
The authors would like to thank the reviewers for their valuable suggestions and sound comments, which have been revised in accordance with the reviewers' comments and have been reasonably explained as follows (with Thermo mechanical coupled contact analysis of alternating meshing gears teeth for marine power rear transmission system considering thermal expansion deformation as an example being discussed).
Marine power rear transmission system high speed large load gears in the process of alternating meshing there is thermal expansion elastic deformation, which causes teeth profile to deviate from standard involute, resulting in strong impact and vibration of gears. The structural coupling system of marine cylindrical gear has obtained by simplifying power rear transmission system into the elastic system affected by thermal expansion and deformation. In order to focus on the influence of marine gears meshing thermal expansion and thermal deformation on contact coupling of power rear transmission system, as shown in Fig. 1, the following basic assumptions are made.
1) The installation of MPRTS is rigid.
2) Only considering contacts with sliding and rolling movements between meshing gears teeth surface.
3) The meshing forces between gears teeth surface are always acting on meshing line.
4) The meshing gear pairs are simplified as a spring damping mass system.
Auxiliary Fig.1 Structural coupling model of MPRTS with thermal expansion elastic
In Auxiliary Fig.1, and are the stiffness coefficients of meshing thermal expansion of driving and driven gears respectively. and are the elastic deformation coefficients of driving and driven gears respectively. and are the driving and driven gears meshing damping coefficients respectively. is the gear meshing impact coefficient. and are the torques of driving and driven gears respectively. and are the torques of driving and driven gears respectively. and are the inertia moments of driving and driven gears respectively. is the gear meshing torque. and are the radii of driving and driven gears respectively. Ignoring the inelastic deformation between gears meshing teeth, the expression can be written as
Auxiliary (1)
In order to highlight the structure contact characteristics of MPRTS, with using generalized coordinates,, , then this expression would be written as
Auxiliary (2)
Considering driving gear meshing thermal expansion stiffness coefficients under the same load sharing excitation, and which making a very significant effect on gear pair impact of coupling contact. In this paper, the Runge-Kutta method with 5-6 order adaptive step sizes have used to solve the Auxiliary equation (2), and the numerical solution of meshing gear coupling contact of MPRTS is obtained. The gear parameters used in the present paper are listed in Auxiliary Table 1. Respectively, as shown in Auxiliary Fig. 2, as shown in Auxiliary Fig. 3, as shown in Auxiliary Fig. 4.
Auxiliary Table 1 The parameters of herringbone gear pairs
Parameter name |
Value |
Number of teeth of driving gear |
36 |
Number of teeth of driven gear |
78 |
Modulus |
6 |
Pressure angle |
20 |
Tooth width |
300 |
Inertia moment of driving gear |
6736 |
Inner ring inertia moment of driven gear |
32000 |
Wheel hub inertia moment of driven gear |
13576 |
Stiffness coefficients of meshing thermal expansion of driving gear |
784924 |
Obtained by numerical analysis that teeth surface structural contact characteristics of meshing region during the whole gear meshing cycle of MPRTS, from these figures can be seen
Auxiliary Fig.2 Input pinion contact characteristics numerical simulation
Auxiliary Fig.3 Input pinion contact characteristics numerical simulation
Auxiliary Fig.4 Input pinion contact characteristics numerical simulation
With the increase of meshing thermal expansion stiffness coefficient, the convergence speed of teeth surface meshing phase is slower. This indicates that meshing thermal expansion stiffness coefficient couldn’t effectively suppress contact impact, but also improve the instability of MPRTS. Therefore, it is imperative to analyze marine meshing gear thermodynamic coupling of power rear transmission system.
Q26. Ll. 275 ff: revise “the integral of the pressure […] dimensionless form as:”
the integral of the pressure multiplied by the microelements contact area in the gear meshing domain equals the total amount of the applied loads [35], which is expressed in dimensionless form as:
Q27. Ll. 282 ff: revise “and the optimized evaluation […] is predicted”
the enhanced homogeneous shared tribological performance of the TEHL oil film with meshing TME interfaces load-bearing capacity are predicted for optimal assessment.
Q28. Eqs. (10) and (12): these separations are unclear; “hybrid regime” means mixed lubrication here,
i.e. partial solid-solid contact? If so, why Reynolds’ equation is still utilized?
The aim of the research in this paper is to develop techniques to improve the micro-geometry of functional MTS as a result of optimized friction behaviour. The deterministic MTS microstructure is a possibility to influence the lubrication regime and, therefore, to adjust friction and wear. In this study, a model based on hydrodynamic lubrication equations for linear slip contact MTS is developed to explain the frictional behaviour of the lubrication-load bearing. The model explains why comparing microtextures with different channel widths, although the model equations are derived from hydrodynamic theory, the microgeometry is responsible for the unusual behaviour and further insight is gained into the mechanisms of MTS microchannels.
Q29. ll. 301 ff: explain “The load-bearing capacity […] to be known”
The authors thank and accept the reasonable suggestions of reviewer 3, and the authors have provided the necessary explanations below.
Hybrid TEHL numerical solution problem is determined by the load conditions that have been set, and the solution pressure must meet the above-mentioned preconditions for load balance. The load balancing equation is expressed as:
Auxiliary (1)
Here, when the parameterexists, denotes the film pressure index of hydrodynamic oil, whereas, is the interface pressure of non-smooth CIMT. For the GTS, is the gear interface DMF of a non-double MGP.
Auxiliary Figure1. Schematic representation of geometric distribution of MGPs composite CIMT at the meshing point
Considering that the friction of the gear meshing interface with TEHL conditions is caused by the hydrodynamic oil film viscous shear stress between the MGP. The aforementioned shear stress is caused by the combination of Poiseuille and Couette flows and varies linearly in real time along the z orientation (that is, along the film thickness orientation of the hydrodynamic oil), and which is denoted as:
Auxiliary (2)
In view of the fact that the meshing interface is not smooth and under the action of external excitation load, the peak position of the rough interface may have uneven and non-smooth contact, thereby forming a hybrid TEHL state. Therefore, the friction force of GTS consists of two parts. One is that there is a hydrodynamic oil film viscous shear stress between the MGP, and the other is that the non-indirect contact of rough peaks leads to the rupture of the oil film on the MGP interface, which weakens the lubrication effect and produces uneven friction thermoelastic behavior. According to the analysis process of hybrid TEHL, the intermediate oil film viscous shear force is regarded as the interface oil film contact friction force, and the contact interface real-time transient friction force of the MGP in any time domain can be shown as:
Auxiliary (3)
Where, and respectively represent the grids number along the and directions in the numerical calculation domain. denotes the grid division zone unit, is the interfacial oil film equivalent viscous transient shear force at the middle interface layer (where in the above Eq. (2)) on mesh nodes (), is the non-smooth interface contact transient pressure on mesh nodes (), and the non-smooth interface contact friction coefficient is assumed to be 0.1. Combined equations (8) and (20), substituted into equation (3), and the contact transient friction force at meshing interface is derived from the following expression:
Auxiliary (4)
Auxiliary (5)
The CIMT friction coefficient mathematical formula is expressed as:
Auxiliary (6)
Based on the film pressure and film thickness (convergent analytical value) of the hybrid TEHL model, the viscous damping, sliding friction force and rolling friction forceof the MGP are analyzed, which can be expressed as follows:
Auxiliary (7)
Auxiliary (8)
Auxiliary (9)
From the aforementioned equation (7), it can be revealed that the lubricating oil viscous damping is proportional to its equivalent viscosity parameter, but inversely proportional to the hydrodynamic oil film thickness at MGP contact interface. The rolling/sliding friction forceincreases with the increase of the viscous damping, the relative slip transient velocity of the MTS and the non-smooth concave-convex contact pressure. The rolling/sliding friction force and the oil film thicknesschange in direct proportion to the oil film pressure gradient of the contact interface along the sliding direction of the meshing teeth pair. The friction equation (4) is calculated from the numerical solution of the hybrid TEHL and substituted into the vibration equation (8). The frictional dynamic coupling equations of the meshing pair of GTS can be expressed as:
Auxiliary (10)
Auxiliary (11)
Auxiliary (12)
Auxiliary (13)
For the above equations (8)-(13), it can be revealed that the rotational and translational DoFs are coupled in equations (10) and (13) along the OLOA direction. Considering that the gear meshing interface frictional force exists in real time, resulting in the interaction between the translational vibration and the torsional vibration of the GTS along the OLOA orientation, the DMF is described as a function of displacementand velocityalong the LOA orientation. The six DoFs are coupled in equations (8)-(11) and are solved jointly by the DMF and frictional forces at the MGP interface. The torsional vibration of GTS and all translational vibrations are in a state of interaction. At each point the three proposed interfacial micro-texture configurations are appraised. The Reynolds and the thermal interface elastohydrodynamic equations are discretized by the finite element method, and the second-order Lagrangian analysis method is adopted. The microscopic problems of the homogenized model are solved in a homogenized manner at each node of the mesh of the macro-problems. However, this results in a large accuracy of degrees of freedom, and since our goal is to evaluate the accuracy of the model proposed here, we want to eliminate sources of error. Those come from the decoupling of macro equations from micro equations. Nonetheless, this will limit our ability to solve minima problems.
Q30. Eq. (14) is unclear; are these different cases or contributions? not all notations are clarified
The authors thank reviewer 3 for his sound advice and for providing the necessary explanation.
A review of the existing literature shows that most of the published work deals with the effect of MTS textures in hydrodynamic regimes. In this study, the effect of MTS textures in the TEHL regime is investigated through experimental simulations and analytical modelling to account for the uneven contact and its contribution to friction. In addition, the behaviour of MTS textures during the wear phase - an effect which, to the best of our knowledge, has not been investigated before, particularly for textured MTS. computational models based on Reynolds equation treatments were developed to predict the performance of textured MTS lubrication contacts at steady state. Experimental simulations were carried out with a pin configuration on a disc. The model considers the effect of MTS microtexturing as the problem is in the TEHL regime.
Q31. ll. 312 ff: revise and explain “Comparing this value […] is evaluated”
The value ofis compared withto determine if the initial assumption of a constant coefficient holds. If this does not hold, another constant factor is assumed and the determination continues until convergence is achieved and the friction factor is evaluated.
Q32. l. 321: “The TME discretization equation”; explain
The authors thank reviewer 3 for his sound advice and for providing the necessary explanation.
This paper uses the Newton-Raphson method to solve the Reynolds equation. This is a faster method compared to the direct method. Another advantage of this method is better convergence at high loads compared to the direct solution method.
A summary of the calculation procedure is given below. The finite difference method is used to discretize the governing equations. The initial pressure distribution (Hertzian distribution) is used as an initial guess to find the lubricant film thickness. A minimum film thickness is then selected and the film thickness is calculated. The film thickness is adjusted by adding the depth of the indentation located in the middle of the contact to the thickness of the lubricant film. The load balance equation, which ensures a balance between the applied load and the hydrodynamic pressure, is used to update the minimum film thickness. All equations are solved simultaneously to predict changes in pressure distribution and film thickness.
Q33. ll. 318 ff: the numerical procedure is still unclear; elaborate, a graphical scheme of the full algorithm is highly desirable
The effect of laser-generated TME on the thermal steady-state behaviour of the MTS is investigated by experimental simulations and numerical analysis using the TEHL model. The experimental simulations are carried out using a pin-disk apparatus. Multi-scale circular indentations are formed in the MTS by laser. The geometry of the dent optimizes the optimum size of the dent to be selected. The pin and the disc form a gear meshing configuration with linear contact Each experiment simulates a line contact pair to investigate the abrasion and thermal steady state behaviour of textured MTS. The height of the action of the dent depends on the velocity, which has a huge impact on the film thickness. Parametric simulation studies are carried out to gain insight into the relationship between input parameters such as velocity, applied load and dent size and film thickness, pressure distribution and load carrying capacity. The model is able to predict the coefficient of friction in the TEHL regime. Comparison of the experimentally obtained friction coefficient values shows good agreement with the results predicted by the computational model.
Q34. Sect. 3: what is the purpose of this section, if no experimental data is used in the manuscript?
The effect of MTS texture on the tribological properties of TEHL wire contacts is investigated through experimental simulations and an analytical computational model. The model predicts the load carrying capacity of untextured and textured MTS. Results are reported from experimental simulations of meshing thermal stability experiments which used a pin-disk test rig to investigate the effects of MTS texturing. This effect is more pronounced at lower loads and higher sliding speeds. These dents, when properly textured, can greatly improve wear behaviour and load-bearing capacity. The model shows that speed plays a major role in shaping film thickness and load-bearing capacity. At low speeds, dents lead to a reduction in film thickness and load-bearing capacity. As speed increases, dents have a positive effect on film thickness and load-bearing capacity.
Q35. Ll. 334 ff: revise the complete passage “The MTS line contact model […] meshing domain”
The MTS line contact model for involute spur gears is experimentally simulated using TME characteristic parameters, which are perpendicular to a selected microtexture interface along the meshing line slip direction, which is periodically and uniformly dis-tributed in the meshing domain.
Q36. L. 356: “The results of […] experimental studies”; experimental data not shown in manuscript
The results of the numerical calculations and experimental simulation studies are reported and discussed in the following presentations.
Q37. ll. 368 f: elaborate on “the effect of thermoelastic […] slip linear velocity”
The authors thank reviewer 3 for his sound advice and have addressed the relevant implications mentioned in accordance with the reviewer's comments. The authors have elaborated the necessary explanations in relation to the results of previous studies.
The structural contact analysis units should be rebuilt after the model that converted into thermal structure. Selecting the structural units’ plane42 and solid45 are regarded to the same nodes as temperature field elements’ plane55 and solid70. Then defining structural material properties and adding thermal expansion coefficients in this analysis. Marine gear alternate meshing area division of power rear transmission system is shown in Auxiliary Fig. 1.
Auxiliary Fig. 1 Marine gear alternate meshing area division of power rear system
By changing meshing position angle, taking the contact stress of 16 meshing positions for analysis within the range of 35 ° of the rotation of driving gear. The parameters of spur gear are shown in Auxiliary Table 1 (Coincidence degree is 1.73). Position 1 meshing-in impact point, rotation angle is 1, meshing joint node number is 224, joint node number of distance from tooth top is 0, as shown in Auxiliary Fig.2. Position 2 double teeth meshing transition to single tooth meshing C point, rotation angle is 15.5, meshing joint node number is 257, joint node number of distance from tooth top is 34, as shown in Auxiliary Fig.3. Position 3 single tooth meshing area, rotation angle is 17.5, meshing joint node number is 254, and joint node number of distance from tooth top is 37, as shown in Auxiliary Fig. 4. Position 4 single tooth meshing transition to double teeth meshing D point, rotation angle is 20.5, meshing joint node number is 250, and joint node number of distance from tooth top is 41, as shown in Auxiliary Fig. 5. Position 5 meshing-out impact point, rotation angle is 35, meshing joint node number is 235, joint node number of distance from tooth top is 55, as shown in Auxiliary Fig. 6.
Auxiliary Table 1 The parameters of spur gear pairs
Teeth number driving gear |
Teeth number driven gear |
Pressure angle |
Modulus |
33 |
51 |
20 |
6 |
Tooth width |
Power |
Input speed |
Initial oil temperature |
330 |
1500 |
2900 |
65 |
Auxiliary Fig. 2 Position 1 meshing-in impact point, rotation angle is 1
Auxiliary Fig. 3 Position 2 double tooth meshing transition to single tooth C point
Auxiliary Fig. 4 Position 3 single tooth meshing area, rotation angle is 17.5
Auxiliary Fig. 5 Position 4 single tooth meshing transition to double teeth D point
Auxiliary Fig. 6 Position 5 meshing-out impact point, rotation angle is 35
In these above figures, obtained that gear deformation amount with different angular at meshing positions, the left side is thermodynamically coupled deformation cloud chart, and the right side is elastic deformation cloud chart. Teeth surface contact stress would be increased rapidly when marine gears are meshed in or meshed out.
Auxiliary Fig. 7 Position 1 meshing-in impact point, rotation angle was 1
Position 1 meshing-in impact point, rotation angle is 1, meshing joint node number is 224, joint node number of distance from tooth top is 0, as shown in Auxiliary Fig. 7.
Auxiliary Fig. 8 Position 2 doubles meshing transition to single C point
Position 2 double teeth meshing transition to single tooth meshing C point, rotation angle is 15.5, meshing joint node number is 257, joint node number of distance from tooth top is 34, as shown in Auxiliary Fig. 8.
Auxiliary Fig. 9 Position 3 single meshing area, rotation angle was 17.5
Position 3 single tooth meshing area, rotation angle is 17.5, meshing joint node number is 254, and joint node number of distance from tooth top is 37, as shown in Auxiliary Fig. 9.
Auxiliary Fig. 10 Position 4 single meshing transition to double D point
Position 4 single tooth meshing transition to double teeth meshing D point, rotation angle is 20.5, meshing joint node number is 250, and joint node number of distance from tooth top is 41, as shown in Auxiliary Fig. 10.
Auxiliary Fig. 11 Position 5 meshing-out impact point, rotation angle 35
Position 5 meshing-out impact point, rotation angle is 35, meshing joint node number is 235, joint node number of distance from tooth top is 55, the contact stress rises when the gear is about to end meshing, as shown in Auxiliary Fig. 11.
In these above figures, obtained that gear contact stress values with different angular at meshing positions, the left side is thermo mechanical coupled contact stress cloud chart, and the right side is elastic contact stress cloud chart.
Auxiliary Fig. 12 Thermo - mechanical coupling deformation and contact stress
Auxiliary Fig. 13 Elastic deformation and elastic contact stress of gears teeth surface
Auxiliary Fig. 14 Driven gear elastic deformation
Auxiliary Fig. 15 Driven gear thermal expansion deformation
Auxiliary Fig. 16 Driving gear thermal expansion elastic deformation
Could be seen from Auxiliary Figures, D is switching point when single tooth meshing area would be transferred to double teeth and above meshing area, it is concerned that the teeth top of driving and driven gears are meshed in and out in alternate meshing area. Thermo - mechanical coupling deformation and contact stress, as shown in Auxiliary Fig. 12. Elastic deformation and elastic contact stress of gears teeth surface, as shown in Auxiliary Fig. 13. Driven gear elastic deformation, as shown in Auxiliary Fig.14. Driven gear thermal expansion deformation, as shown in Auxiliary Fig. 15. Driving gear thermal expansion elastic deformation is shown in Auxiliary Fig. 16. Due to the influence of temperature, meshing stiffness and teeth shape, the maximum amount of thermal expansion deformation during gear meshing occurs at the teeth top.
Thermal expansion leads to the increase of gear tooth thickness, with comparing to standard teeth profile; the teeth height and entire teeth shape after thermal expansion are relatively larger. In Auxiliary Fig. 16, the maximum elastic deformation amount of driving gear at the critical point with single and double teeth alternating meshing is 44 microns. And the value is never to be ignored in MPRTS actual operating conditions.
Q38. Figs. 6 through 11 as well as Fig. 4 are each given twice
The authors acknowledge and accept the sound advice given by the reviewers and have revised the manuscript.
Q39. l. 425: revise “without microtextured”
The authors have revised "without microtextured" to "non-microtextured". A total of two have been made in this manuscript.
Q40. ll. 427 ff: average oil film thickness is not shown
In this case, the minimum film thickness of the untextured MTS is greater than that of a textured MTS with different values of dimple depth. Increasing the dimple aspect ratio leads to a reduction in the average film thickness. The average film thickness in the untextured case is less than the average film thickness of the textured MTS. The average film thickness is calculated over the entire contact area and it is the minimum film thickness which is one of the key parameters determining the tribological performance of textured MTS. Protrusions with a height greater than the minimum film thickness produce greater friction, heat and wear particles, although the height of these protrusions may be less than the average film thickness.
Q41. Fig. 10: axis says “minimum contact pressure”
One of the main objectives of MTS texturing is to increase load-bearing capacity. Textured MTS has a thicker average lubricant film than non-textured MTS and the minimum contact pressure is distributed over a larger area, with two pressure peaks occurring at the edges of the dimples. When the aspect ratio is low, the area with a constant area is wider and a larger pressure peak occurs at the trailing edge of the dimple to meet the load balance. This larger pressure peak causes the film thickness to drop to a lower level than the minimum film thickness for the same conditions of the untextured MTS.
Q42. ll. 449 f: revise “More interestingly […] contact pressure.”
The authors have revised and reflected in the paper following the comments of Reviewer 3.
Q43. Fig. 11: how is the pressure normalized? also, the differences between different TME are very small; how can the authors substantiate that these are significant compared to the margin of error of the numerical model?
The maximum pressure variation as a function of crater geometry was compared for different textured and non-textured samples. The results show that the peak pressure of the textured MTS is greater than that of the non-textured MTS, especially for pits with low aspect ratios. Lower crater aspect ratios result in wider craters and a more even pressure distribution. Therefore, load balancing should be satisfied by higher peak pressures.
It is worth noting that the variation in maximum pressure with nest depth shown in the graph does not necessarily mean that producing deeper nests will result in better tribological performance. The variation of pressure with different nest depths is illustrated in the graph. The area under the pressure curve shows the load-bearing capacity. The results show that shallower dents are more effective.
Q44. results for the macroscopic coefficient of friction are not shown
In a follow-up study, to check the repeatability of the tests, a number of experiments were repeated for both textured and untextured MTS to show good repeatability of the friction coefficients. Comparison of the friction coefficients obtained from simulations and experiments for the untextured MTS. The experimental results of the simulation data to predict the coefficient of friction are obtained at different speeds. The expected result, due to the increase in lubricant film thickness, is that the friction decreases with increasing speed. The experimental results and simulated friction coefficients are compared with the variation of speed in the case of textured MTS. In the experimental results friction coefficients are obtained at different velocities. The numerically calculated best textured samples show better agreement compared to the simulated results.
Q45. thermoelastic effects are basically not discussed at all
Numerous reported and studied results show that microfabrication techniques do significantly improve the friction reduction and load-bearing capacity of the interface of contact parts. The multiscale geometrical parameters of microtexturing are one of the central factors in optimizing the benefits of microtexturing interfaces. It is emphasized that these known results provide significant improvements in terms of brief frictional wear and lubricant load-bearing. In most cases, the contact interface microfabrication is presented in a pre-engineered form. Undoubtedly, each complexity parameter correlation and mechanistic understanding is not yet fully understood, and the analytical descriptions are only in a limited number of hypothetical cases, failing to break through most of the known mechanisms limited to discrete textures of crater area ratios and pattern shapes, and furthermore, little pre-research work has been done on multi-scale contact interface textures from the perspective of geometric parameter characterization.
Assuming that multiscale weaving interfaces are microhomogeneous and periodic, a formal approach based on the decoupling of macroscale and microscale is introduced to introduce a micro-element homogenised interface micro TEHL dynamic pressure contact model that considers the non-negligible deformation due to compressive load bearing and thermal effects at the microscale, which extends the general applicability of the classical progressive homogenization approach. The correlative effect of MTS texturization on the tribological properties of TEHL slip linear contact is investigated by means of simulated experiments and numerical computational models, which are used to assess the friction coefficients of microtextured MTS, and to analyze and predict the load-bearing capacity of MTS with and without interfacial TME features.
The search for correlation between key optimal micro-element shape design parameters and geometric scale parameters is necessary to obtain maximum frictional wear reduction benefits and thus improve performance and pursue this concept for lubricated load-bearing engineering applications at gear meshing interfaces. In this study, pre-designed micro-element weaving patterns of slip line contact interfaces under thermoelastic flow lubrication conditions reveal the correlation between friction and load-bearing capacity of key parameters such as depth and width under mixed thermoelastic hydrodynamic lubrication.
Q46. Sect. 5: in the discussion, the authors need to elaborate on the following questions: How are the obtained results validated without experimental data? What are the drawbacks of the present study (e.g., with respect to homogenization and decoupling of scales)?
It is known from the simulations that increasing the slip line speed leads to an easier formation of a thicker TEHL lubrication film, which in turn affects the location and magnitude of the contact pressure spikes. The pressure distribution and TEHL film thickness at the microtexture meshing interface are elucidated for the same applied load and slip line speed. the presence of the TME feature results in two contact pressure peaks at its edges and a flat and relatively low pressure contact region in the middle of the meshing area. Note that the reduction in TEHL film thickness corresponding to the contact pressure peaks is expected to be even lower than the minimum TEHL film thickness at the untextured engagement interface. However, the average oil film thickness of the MTS is increased by the technical means of texturing the contact interface. An interesting finding is that the minimum TEHL film thickness of the MTS occurs at the leading edge of the TME boundary.
The minimum TEHL oil film thickness and the average oil film thickness of the microtexture MTS are discussed in detail in the experimental simulations under the same applied load and slip linear velocity conditions. The results of the analysis show that increasing the interface TME aspect ratio leads to an increase in the minimum TEHL film thickness value. In this case, the minimum TEHL film thickness of the non- microtextured MTS is greater than the minimum TEHL film thickness of the microtextured MTS with different TME depth values. Increasing the interfacial TME aspect ratio leads to a decrease in the average MTS film thickness. The average film thickness without microtexture is less than the average film thickness on the microtexture MTS. The average film thickness is calculated over the entire contact area and the minimum TEHL film thickness is one of the key parameters determining the tribological performance of the microtextured MTS. Micro-bumps with a height greater than the TME minimum TEHL film thickness will generate greater friction, heat and wear particles, even though the height of these micro-bumps may be less than the average MTS film thickness.
Q47. ll. 482 ff: revise “Simulation-based mating […] is investigated.”
Mating interfaces based on simulated microtextured MTS lubrication performance - friction/wear behaviour - ATSLB capability concentrated conformal slip wire contact are investigated using TEHL formulations.
Contribution of each individual co-author:
No conflict of interest exits in the submission of this manuscript, and manuscript is approved by all authors for publication. We would like to declare on behalf of our co-authors that the work described was original research that has not been published previously, and not under consideration for publication elsewhere, in whole or in part. All the authors listed have approved the manuscript that is enclosed.
In this subject research, we have proposed the “Numerical Analysis of Friction Reduction and ATSLB Capacity of Lubricated MTS with Textured Micro-Elements” for a theoretical model of thermoelastic lubricated interfacial Textured Micro-Element (TME) load-bearing contact, and the effective friction reduction and Anti-Thermoelastic Scuffing Load-Bearing (ATSLB) characteristics between random rough Meshing Teeth Surfaces (MTS) are investigated, the mechanism linking interfacial thermoelastic lubrication, TME meshing friction reduction and ATSLB is revealed. The MTS is pre-set as a line contact mode, which breaks through the limitation of assuming that the actual meshing area is much smaller than the nominal interface contact domain. The real contact domain area between MTS with multi-scale Micro-Element Textures (MET) is obtained for the numerical calculation of the three-dimensional equivalent TME contact volume, which is the correlation bridge between friction reduction and ATSLB of the thermoelastic lubrication interface.
Our main contribution to the field is to predict the time-varying behaviour of the textured meshing interface friction reduction with TME contact load under thermoelastic lubrication conditions by the proposed theoretical model. Numerical simulations show that the textured interface meshing volume is the key to solving the load-bearing problem of line contact between randomly rough teeth surfaces. The friction coefficients of the MTS are reduced by 13-24%. Interfacial MET parameters are optimized and the correlation between linear velocity, time-varying load and micro-configuration scale and meshing interface oil film thickness, oil film pressure distribution morphological trends and ATSLB capacity is elaborated. The lubricated load-bearing and friction reduction behaviour between the textured MTS is quantified by the thermoelastic voids of the TME interface and the actual meshing volume ratio, which provides a new perspective for further insight into the lubrication and friction reduction behaviour between the MTS with multi-scale MET-ATSLB coupling mechanism.
The novelty and significance of this manuscript are as follows:
The work was studied mainly “Numerical Analysis of Friction Reduction and ATSLB Capacity of Lubricated MTS with Textured Micro-Elements”. This topic involves the homogenized micro hydrodynamics of the MTS with interfacial TME characteristics in the TEHL steady state is approximated accurately, the microgeometry of the multiscale structure with contact interface TME features is simulated, the existence of an optimal geometry that is particularly effective for the load-bearing capacity of the MTS is proposed and discussed, and the validity of the optimized MTS with interfacial TME features significantly affecting the load-bearing capacity of the macroscopic properties of the generic slip linear contact is confirmed.
We are very hoped to publish this article in your journal, and I thank you on behalf of our group. We apologize for what we have not done well. We hope we will continue to submit better articles to you.
Thank you in advance for considering this revised submission. We very much look forward to your reply and any questions needed next.
Thank you and best regards.
Sincerely,
Corresponding author/First author
Xigui Wang Professor, PhD Supervisor
School of Engineering Technology, Northeast Forestry University, No. 26, Hexing Road, Xiangfang District, Harbin, 150040, China; School of Mechatronics and Automation, Huaqiao University, No. 668 Jimei Avenue, Jimei District, Fujian Province, Xiamen, 361021, China
Hui Huang
School of Mechatronics and Automation, Huaqiao University, No. 668 Jimei Avenue, Jimei District, Fujian Province, Xiamen, 361021, China;
Jingyu Song
School of Mechatronics and Automation, Huaqiao University, No. 668 Jimei Avenue, Jimei District, Fujian Province, Xiamen, 361021, China;
Yongmei Wang
School of Motorcar Engineering, Heilongjiang Institute of Technology, No. 999, Hongqidajie Road, Daowai District, Harbin, 150036, China; School of Mechatronics and Automation, Huaqiao University, No. 668 Jimei Avenue, Jimei District, Fujian Province, Xiamen, 361021, China
Jiafu Ruan
School of Engineering Technology, Northeast Forestry University, No. 26, Hexing Road, Xiangfang District, Harbin, 150040, China; School of Mechatronics and Automation, Huaqiao University, No. 668 Jimei Avenue, Jimei District, Fujian Province, Xiamen, 361021, China

Round 4
Reviewer 1 Report
Please see the attached review file.

Author Response
February 2, 2023
Dear editor-in-chief
The open access journal Lubricants
We have submitted a research article hoping to be published in the journal Lubricants, titled “Numerical Analysis of Friction Reduction and ATSLB Capacity of Lubricated MTS with Textured Micro-Elements.” The paper was coauthored by XiguiWang, Hui Huang, Jingyu Song, Yongmei Wang, and Jiafu Ruan. We have checked the manuscript and revised it according to the comments. Overall, the comments have been fair, encouraging and constructive. We have learned much from it. We submit here the revised manuscript to meet the evaluation conditions and requirements of the reviewers.
Response to Reviewers' Comments:
Reviewer 1:
On behalf of all members of our team, I would like to thank the Reviewer 2 for approving and accepting this manuscript. The authors have once again fully reviewed and refined this revised manuscript, and reflected the refined content in this article. In the long-term in the future, our research team will submit more excellent and high-quality articles to your journal.
Q1. Answer to Q4: “original source” refers to the original source from 1966, which should be included, if possible.
Sincerely respond to Q1 raised by Reviewer 1:
The authors have made revisions as suggested by Reviewer 1 and reflected in this manuscript.
Original Reference 4:
- Zhang H., Liu Y., Hua M., Zhang D. Y., Qin L. G., Dong G. N. An optimization research on the coverage of micro-textures arranged on bearing sliders. Tribology International 2018, Vol. 128, p. 231-239.
Revised Reference 4:
- Hamilton D. B., Walowit J. A., Allen C. M. A Theory of Lubrication by Microirregularities. Journal of Basic Engineering 1966, Vol. 88, Issues 1, p. 177-185(9 pages).
Q2. Answer to Q5: “while also is just an unveiling” still seems grammatically incorrect.
Sincerely respond to Q2 raised by Reviewer 1:
The authors have made English polishing revisions to this manuscript as suggested by Reviewer 1, and these revisions are reflected in this manuscript.
Revised as follows:
"while also is just an unveiling" is hereby revised to "and is also only an unveiling".
Q3. Answer to Q8: revise “patterns of microtextured” and “the experimental results […] that the friction coefficient […]”.
Sincerely respond to Q3 raised by Reviewer 1:
The authors have made English polishing revisions to this manuscript as suggested by Reviewer 1, and these revisions are reflected in this manuscript.
Revised as follows:
"patterns of microtextured” and “the experimental results […] that the friction coefficient […]" is hereby revised to "the experimental results illustrated by the Stribeck curves show that in the EHL state, the coefficient of friction at the interface of the squared micro-textured holes is re-duced by approximately 6% and the coefficient of friction at the interface of the parallel micro-element textured grooves is increased by approximately 81% in the hybrid TEHL state [14-16]. A theoretical model for studying the laser interface microtexture topography pattern of non-rigid elastomers in EHL contacts is proposed, and the Reynolds equation and the elasticity equation of non-rigid elastomers are solved. These solutions show that the desired reduction of the contact friction coefficient by 30% can be achieved under optimum lubrication conditions at the interface [17-19]".
Q4. Answer to Q11: “lubrication performance-friction/wear behavior-load bearing capacity” is still difficult to read; consider separating by words, instead of hyphens
Sincerely respond to Q4 raised by Reviewer 1:
The authors have made English polishing revisions to this manuscript as suggested by Reviewer 1, and these revisions are reflected in this manuscript.
Revised as follows:
"lubrication performance-friction/wear behavior-load bearing capacity" is hereby revised to "lubrication performance, frictional wear behaviour and load bearing capacity".
Q5. Answer to Q13: see previous comment regarding Q11
Sincerely respond to Q5 raised by Reviewer 1:
The authors have made English polishing revisions to this manuscript as suggested by Reviewer 1, and these revisions are reflected in this manuscript.
Revised as follows:
"lubrication performance-friction/wear behavior-load bearing capacity" is hereby revised to "lubrication performance, frictional wear behaviour and load bearing capacity".
Q6. Answer to Q16: revise “analytical descriptions are only in a confined number of hypothetical cases” and “crater”; consider starting a new sentence at “Furthermore, […]”
Sincerely respond to Q6 raised by Reviewer 1:
The authors have made English polishing revisions to this manuscript as suggested by Reviewer 1, and these revisions are reflected in this manuscript.
Revised as follows:
"the analytical descriptions are only in a confined number of hypothetical cases and failing to break through most of the known mechanisms, being limited to crater area ratios and patterned shapes of discrete microelements textures," is hereby revised to "the analysis describes only a limited number of hypothetical cases that fail to break through most of the known mechanisms and is limited to crater area ratios and pattern shapes of discrete micro-element textures."
Q7. Answer to Q21: “U represents the scale factor” is unclear”
Sincerely respond to Q7 raised by Reviewer 1:
Revisions have been made by the authors in response to the suggestions of reviewer 1 and are reflected in this manuscript. The authors have provided the necessary explanations and clarifications here below.
The film thickness is calculated from the mean gap after deformation, flow and roughness contact are treated in a unified model. In the hydrodynamic lubrication region, the pressure is controlled by the Reynolds equation expressed as follows.
where the x coordinate coincides with the direction of motion and .
Q8. Answer to Q22: in the manuscript, “a” is still not in italics
Sincerely respond to Q8 raised by Reviewer 1:
Revisions have been made by the authors in response to the suggestions of reviewer 1 and are reflected in this manuscript. The authors have provided revised examples here to illustrate this.
Here, a denotes half-width of the MTS contact region,
Q9. Answer to Q24: I guess, what the authors give in the cover letter is a copy from their previous work; for the present manuscript to be comprehensible, this previous work must be referenced more clearly, and a brief summary of the procedure should be given in the manuscript
Sincerely respond to Q9 raised by Reviewer 1:
The authors acknowledge and accept the reasonable suggestions and valuable comments made by the reviewers, who have sorted through previous research work and cited it appropriately in this manuscript and summarized it succinctly.
One of the examples of added and revised content:
The numerical generation process and the hybrid TEHL model for "'microfabricated' MTS constitute a virtual analysis system, which is used to create, study and compare a series of textured MTS, and perform numerical experiment simulation to verify and evaluate the TEHL performance of the generated MTS.
Q10. Answer to Q25: That does not answer my previous comment at all, which is about the meaning/derivation of explicitly Eq. (4)
Sincerely respond to Q10 raised by Reviewer 1:
The authors acknowledge and accept the reasonable suggestions made by the reviewers, and the authors have provided the necessary explanation below in response to questions about the meaning/derivation of the explicit formula (4).
Previous studies in the group have reported that the convergence of the teeth meshing phase slows down as the meshing thermal expansion stiffness factor increases. This suggests that the meshing thermal expansion stiffness factor is not effective in suppressing contact shock, but also improves the instability of the MTS line contact lubrication performance, frictional wear behaviour and load bearing capacity.
In order to find an acceleration of the MTS thermoelastic deformation calculation, which is a major part of the total solution process, a Discrete Convolution-Fast Fourier Transform method has been reported in previous studies and this has been incorporated into the TEHL model presented in this paper, for which the calculation has been shown to be fast.
In the gear meshing region, the model results show a continuous and smooth transition to TEHL contact as the line contact slip velocity is continuously reduced. The current hybrid TEHL contact model has proved to be successful in linking lubrication performance, frictional wear behaviour and load bearing capacity and is valid for this study.
Q11. Answer to Q28: That also does not address my questions and concerns at all.
Q28. Eqs. (10) and (12): these separations are unclear; “hybrid regime” means mixed lubrication here, i.e. partial solid-solid contact? If so, why Reynolds’ equation is still utilized?
Sincerely respond to Q11 raised by Reviewer 1:
The authors would like to thank the reviewers for their valuable suggestions and sound comments. Here again, the authors have carefully responded to the issues involved and have provided the necessary explanations below.
Mixed lubrication is a common combination of boundary lubrication, thin film lubrication, microscopic elastohydrodynamic lubrication, hydrodynamic lubrication, etc. All kinds of lubricating films are mainly characterized by film thickness, and their formation mechanism, lubricity and failure criteria are different.
The overall characteristics of mixed lubrication are the comprehensive performance of various lubricant film composition characteristics. The proportion of various lubricating films on the contact surface is related to the shape of the friction interface and working conditions. During the friction process, the proportion and distribution of various lubricating films are constantly changing, so the mixed lubrication characteristics have strong time-varying properties (reported by the research group in previous published discussions). The important feature of MTS mixed lubrication is that it is accompanied by MTS wear, and the main forms are contact fatigue mechanism and adhesion mechanism. According to different environmental media and working conditions, the primary and secondary of these two types of wear mechanisms are different. In the MTS slip line contact friction behaviour thermal effect effects, the MTS interface layer material is subjected to dynamic stress effects and leads to elasto-plastic deformation, while the stress, strain and its bearing volume and material strength are time-varying. Normal wear under mixed lubrication in MTS is mainly due to contact fatigue resulting from dynamic stress fields and microcutting derived from the movement of the abrasive grains. The thermal effects of friction are characterised by parameters such as contact flash temperature, surface temperature distribution and temperature gradients along the depth direction, which are fundamental factors in determining lubricant film failure and adhesive wear, and which are also time-varying.
Q12. Answer to Q29: No aspect of the very long answer (I guess, this, again, is a copy of previous work) is given in the revised version of the manuscript. Also, the comment should be answered briefly and not on two pages.
Q29. ll. 301 ff: explain “The load-bearing capacity […] to be known”
Sincerely respond to Q12 raised by Reviewer 1:
The authors are very grateful to the reviewers for their valuable suggestions and sound comments. Here again, the authors have responded carefully to the issues addressed and have provided the necessary explanations below.
Considering that the meshing interface is not smooth, under the action of external excitation load, the peak position of the rough interface may appear uneven and non-smooth contact, thus forming a mixed TEHL state. The load-bearing capacity of MTS is manifested. One is the presence of hydrodynamic oil film viscous shear stresses between the MTS, and the other is that the non-direct contact of the rough peaks leads to oil film rupture at the MTS contact interface, weakening the lubrication load-bearing effect and producing uneven frictional thermoelastic behaviour.
The load-bearing capacity of the MTS is correlated to the assessment of the microtexture configuration of the MTS contact interface. The Reynolds and thermal interface elastohydrodynamic equations are discretized using a finite element method with a second order Lagrangian analysis. The microscopic problem of the homogenised model is solved in a homogeneous manner at each node of the mesh of the macroscopic problem. This leads to a large degree of freedom accuracy and the objective is to assess the accuracy of the model presented in this paper and hopefully eliminate sources of error. These come from the decoupling of the macroscopic equations from the microscopic equations. These issues mentioned above are known studies that have been explored by this group is the past.
Q13. Answer to Q30: That, again, is not an answer to my questions.
Q30. Eq. (14) is unclear; are these different cases or contributions? not all notations are clarified
Sincerely respond to Q13 raised by Reviewer 1:
The authors are very grateful to the reviewers for their valuable suggestions and sound comments. Here again, the authors have responded carefully to the issues addressed and have added revisions to this manuscript to explain the meaning of the relevant parameters.
Equation (14) evaluates the oil film thickness of the MTS in a mixed lubrication condition, taking into account film thickness values, where rough spots in contact with the MTS may experience elastic, elasto-plastic or plastic deformation situations, which is a contribution of this paper.
where denotes the load carried by TME , is the load carried by a single radius of the TME (indentation), is the load carried by the mean radius of the micro-concave peak tip, is the load carried by the contact area of the TME interface. identifies the equivalent elastic modulus, is the mean radius of the micro-concave peak tip, is the Textured Micro-Element (TME) (indentation) of TME j, is the TME (indentation) of TME k, is the TME (indentation) of TME l, is the interface contact area of a TME in the TEHL regime, and is the gear material hardness. The total forces subjects to the calculation of the load carried by the TME, . The value of is compared with to determine if the initial assumption of a constant coefficient holds. If this does not hold, another constant factor is assumed and the determination continues until convergence is achieved and the friction factor is evaluated.
Q14. Answer to Q32: Again, not answering my very concrete comment: What is the “discretization equation”?
Q32. l. 321: “The TME discretization equation”; explain
Sincerely respond to Q14 raised by Reviewer 1:
The authors are very grateful to the reviewers for their valuable suggestions and sound comments. Here again, the authors have carefully responded to and explained the issues covered.
An outline of the computational procedure in this paper is given below. The finite difference method is used to discretize the control equations. The initial pressure distribution (Hertzian distribution) is used as an initial guess to find the thickness of the lubricant film. The minimum film thickness is then selected and the film thickness is calculated. The film thickness is adjusted by adding the depth of the crater located in the middle of the contact to the lubricant film thickness. The load balance equation, which ensures a balance between the applied load and the dynamic fluid pressure, is used to update the minimum film thickness. All equations are solved simultaneously to predict the pressure distribution and film thickness variation.
Q15. Answer to Q33: Not addressing my comment. Please provide a flowchart of the full algorithm of the numerical model.
Q33. ll. 318 ff: the numerical procedure is still unclear; elaborate, a graphical scheme of the full algorithm is highly desirable
Sincerely respond to Q15 raised by Reviewer 1:
The authors have accepted and greatly appreciate the suggestions made by the reviewers, and the authors have provided a full algorithmic flowchart of the numerical model and have reflected these revisions in this manuscript.
Revised contents have been added as follows:
Algorithm flow chart of multi-scale numerical model algorithm for lubricated MTS with TME considering ATSLB capacity
Q16. Answer to Q34: Not addressing my comment. What is the methodological meaning of a section to describe an experimental apparatus, if no experimental data is used in the manuscript?
Q34. Sect. 3: what is the purpose of this section, if no experimental data is used in the manuscript?
Sincerely respond to Q16 raised by Reviewer 1:
The authors accept and are very grateful to the reviewers for their suggestions, and the authors have provided explanations for the issues raised by the reviewers.
A pin-on-disk test apparatus is used to experimentally study the effect of different parameters on the contact of a pin and the textured MTS under lubricated conditions.
In Section 4, the results of the numerical simulations, as well as an experimental study, are reported and discussed.
For the research purpose of this experimental simulation, increasing the MTS line contact slip velocity will lead to the formation of a thicker lubricating oil film with greater load-carrying capacity, and it affects the position and size of the pressure peak. This is the original intention of this experimental simulation verification.
The relevant Figures in this manuscript show the pressure distribution and film thickness of the textured MTS for the same applied load and velocity. The presence of a dent produces two pressure peaks at the edges of the dent, with a flat and relatively low pressure region in the middle of the contact area. This reduction in film thickness corresponding to the pressure peaks is predicted to be even lower than the minimum film thickness of the untextured MTS.
Q17. Answer to Q37: That answer is not really relevant for my comment; it should be detailed briefly in the manuscript, how it can be seen that “the effect of thermoelastic deformation on meshing interface (sic!) is negligible under a higher slip linear velocity” and how that is relevant for the investigation.
Q37. ll. 368 f: elaborate on “the effect of thermoelastic […] slip linear velocity”
Sincerely respond to Q17 raised by Reviewer 1:
The authors accept and are very grateful for the reviewers' suggestions and the authors have provided explanations for the issues raised by the reviewers.
The authors' intended meaning of the phrase " the effect of thermoelastic deformation on meshing interface is negligible under a higher slip linear velocity " is as follows:
The effect of thermoelastic deformation on meshing interface tends to be overlooked during the numerical simulations and simulated experiments in this paper for the low speed and light load conditions involved in the discussion of the lubrication-load correlation of finite length line contact MTS.
Revised statement:
and the effect of thermoelastic deformation on the meshing interface deserves high attention in future research, especially for high speed, heavy duty MTS.
In the last reply to your comments, combined with a view to the relevant issues studied by this group to express, compared with the standard tooth shape, thermal expansion leads to an increase in gear tooth thickness, thermal expansion of the tooth height and the entire tooth shape is relatively large, especially single and double tooth alternating meshing critical point active gear thermoelastic deformation is the largest, and thus want to express the actual operating conditions in high-speed heavy load infinitely long line contact MTS, this value is not negligible.
This research project uses a Fast Fourier Transform (FFT) based method to calculate the thermoelastic deformation of the MTS. Furthermore, the contact loads between the TMEs are considered to be in the form of point-loading conditions. Consequently, the thermoelastic deformation of each contact node can be calculated by using the following equation. The parameter, , is then used to modify the local film thickness:
where is an influential coefficient at nodal point ; is the local thermoelastic deformation at nodal point ; is the local contact/hydrodynamic pressure at nodal point and having unit of MPa.
Q18. Answer to Q43: Again, not answering my question: How, i.e., using which quantity, was the pressure normalized to obtain nondimensional values?
Q43. Fig. 11: how is the pressure normalized? also, the differences between different TME are very small; how can the authors substantiate that these are significant compared to the margin of error of the numerical model?
Sincerely respond to Q18 raised by Reviewer 1:
The authors accept and are very grateful for the reviewers' suggestions and have explained the issues raised by the reviewers with a view to obtaining your approval.
The numerical calculation of the differences between the different TMEs in relation to the error of the numerical model was carried out by using the models and equations mentioned and presented in this paper. The sequence of pressure normalization to obtain non-dimensional values for the calculations essentially involves the following as described below. (1) setting initial parameters such as pressure distribution, film thickness, etc.; (2) calculating the hydrodynamic pressure distribution; and (3) checking whether the local film thickness is positive or negative. If the local film thickness is negative, calculate the contact pressure, otherwise set the contact pressure to zero; (4) calculate the elastic deformation of the substrate and update the new distribution of dimensionless film thickness; (5) compare the new hydrodynamic pressure distribution with its old counterpart to check convergence; (6) output the result if convergence is complete, otherwise repeat the calculation cycle from step (2) to step (5) until the pressure converges to the allowed error accuracy.
Calculating the film thickness for a given/known load requires another additional calculation cycle to correctly modify the film thickness and update the load to the correct iterative value until the error between the obtained load and its initial given/known value is below or within an allowable accuracy.
Q19. Answer to Q44: A summarized version of this answer should be added to the manuscript.
Q44. results for the macroscopic coefficient of friction are not shown
Sincerely respond to Q19 raised by Reviewer 1:
The authors accept and are very grateful for the reviewers' suggestions and have provided a summarized version of the issues raised by the reviewers, which have been reflected in this manuscript in revised form.
The simulated data predicting the coefficient of friction are obtained at different speeds. The expected result is that the friction decreases with increasing speed for increasing lubricant film thickness. The simulated coefficient of friction is compared with the variation in speed in the presence of a textured MTS. The simulated friction coefficients are obtained at different speeds. The numerically calculated best textured samples show a better agreement.
Q20. Answer to Q45: Not addressing my comment.
Q45. thermoelastic effects are basically not discussed at all
Sincerely respond to Q20 raised by Reviewer 1:
The authors accept and are very grateful for the reviewers' suggestions and the authors have provided explanations for the issues raised by the reviewers.
(1) This paper assumes that the multi-scale weaving interface is micro-homogeneous and periodic. On the basis of the decoupling of macro-scale and micro-scale, a micro-TEHL dynamic pressure contact model of micro-element homogenization interface is introduced. Non-negligible deformations on the scale due to compressive load bearing and thermal effects extend the general applicability of the classical progressive homogenization method. The relevant effects of MTS texturing on the tribological properties of TEHL sliding linear contacts were investigated by simulation experiments and numerical computational models, which were used to evaluate the friction coefficient of microtextured MTS, and to analyze and predict MTS with and without interfacial TME features carrying capacity.
(2) The core objective of this paper is to find the correlation between key optimum micro-element shape design parameters and geometric scale parameters necessary to obtain maximum frictional wear reduction benefits and thus improved performance, and to pursue this concept for lubricated load bearing engineering applications at TEHL gear meshing interfaces.
(3) In this study, pre-designed micro-element texture patterns of sliding line contact interfaces under thermoelastic hydrodynamic lubrication conditions reveal the correlation between friction and load-bearing for key parameters such as depth and width under mixed thermoelastic hydrodynamic lubrication conditions.
Q21. Answer to Q46: Not addressing my comment.
Q46. Sect. 5: in the discussion, the authors need to elaborate on the following questions: How are the obtained results validated without experimental data? What are the drawbacks of the present study (e.g., with respect to homogenization and decoupling of scales)?
Sincerely respond to Q21 raised by Reviewer 1:
The authors have provided explanations for the issues raised by the reviewers.
This paper combines the TME scale parameter characteristics of the line contact interface and the actual requirements of the ATSLB capability of the meshing tooth surface, constructs a dynamic homogenized micro-elastic flow lubrication model for line contact roll/slip friction, investigates the mechanism of the TME scale parameters and cloth state characteristics of the line contact interface, reveals the influence law of the TME scale boundary and the characterization of the frictional thermo-elastic properties of the meshing tooth surface, and establishes a TME scale parameter accurate characterization method in a physically realizable sense. The relationship between the TME function and characteristics of the line contact interface and the variation of the ATSLB capacity of the meshing tooth surface under TEHL condition is explored; with the help of nonlinear strong coupling field solution, the influence of the TME multi-scale parameters on the ATSLB capacity of the meshing tooth surface is resolved by considering the non-Newtonian characteristics of the lubricating medium between the interfaces under heavy load meshing condition. To obtain the optimum distribution interval of TME multi-scale structure parameters for the interface of the meshing tooth table when the oil film properties are in good condition during the high strength contact; to adopt the multiple mesh method and iterate the coarse/fine mesh to quickly eliminate the error to achieve convergence, to analyze the variation characteristics of IEL oil film and load bearing characteristics under different TME distribution density, offset distance and transverse to longitudinal spacing ratio, and to optimize the TME related parameters to achieve the effect on The TME-related parameters are optimized to achieve synergistic regulation of the improvement of ATSLB capability and load carrying capacity of the meshed tooth surface.
In this paper, an analytical numerical model of MTS texture is established. The model contributes to the understanding of tribological behavior in mixed lubrication regimes on textured surfaces, allowing prediction of hydrodynamic pressure and contact pressure between textured MTS, using the mean flow Reynolds equation and TEHL model, respectively. In addition, the FFT method is employed to calculate the thermoelastic deformation of the MTS. The validity of this mixed lubrication model is confirmed by comparing the predictions of the current model with the experimental results of the dial. The predictions of this model allow to obtain the characteristic curve of the MTS texture, which shows that the minimum friction coefficient occurs in the mixed lubrication system and is quite close to the mixed lubrication transition point of the tribological system. The simulation results demonstrate the reduction of friction and wear of the MTS texture. The main mechanism of this effect may be attributed to the hydrodynamic force of the MTS texture to enhance its load-carrying capacity, increase the fit clearance, and reduce the probability of MTS contact.
Contribution of each individual co-author:
No conflict of interest exits in the submission of this manuscript, and manuscript is approved by all authors for publication. We would like to declare on behalf of our co-authors that the work described was original research that has not been published previously, and not under consideration for publication elsewhere, in whole or in part. All the authors listed have approved the manuscript that is enclosed.
In this subject research, we have proposed the “Numerical Analysis of Friction Reduction and ATSLB Capacity of Lubricated MTS with Textured Micro-Elements” for a theoretical model of thermoelastic lubricated interfacial Textured Micro-Element (TME) load-bearing contact, and the effective friction reduction and Anti-Thermoelastic Scuffing Load-Bearing (ATSLB) characteristics between random rough Meshing Teeth Surfaces (MTS) are investigated, the mechanism linking interfacial thermoelastic lubrication, TME meshing friction reduction and ATSLB is revealed. The MTS is pre-set as a line contact mode, which breaks through the limitation of assuming that the actual meshing area is much smaller than the nominal interface contact domain. The real contact domain area between MTS with multi-scale Micro-Element Textures (MET) is obtained for the numerical calculation of the three-dimensional equivalent TME contact volume, which is the correlation bridge between friction reduction and ATSLB of the thermoelastic lubrication interface.
Our main contribution to the field is to predict the time-varying behaviour of the textured meshing interface friction reduction with TME contact load under thermoelastic lubrication conditions by the proposed theoretical model. Numerical simulations show that the textured interface meshing volume is the key to solving the load-bearing problem of line contact between randomly rough teeth surfaces. The friction coefficients of the MTS are reduced by 13-24%. Interfacial MET parameters are optimized and the correlation between linear velocity, time-varying load and micro-configuration scale and meshing interface oil film thickness, oil film pressure distribution morphological trends and ATSLB capacity is elaborated. The lubricated load-bearing and friction reduction behaviour between the textured MTS is quantified by the thermoelastic voids of the TME interface and the actual meshing volume ratio, which provides a new perspective for further insight into the lubrication and friction reduction behaviour between the MTS with multi-scale MET-ATSLB coupling mechanism.
The novelty and significance of this manuscript are as follows:
The work was studied mainly “Numerical Analysis of Friction Reduction and ATSLB Capacity of Lubricated MTS with Textured Micro-Elements”. This topic involves the homogenized micro hydrodynamics of the MTS with interfacial TME characteristics in the TEHL steady state is approximated accurately, the microgeometry of the multiscale structure with contact interface TME features is simulated, the existence of an optimal geometry that is particularly effective for the load-bearing capacity of the MTS is proposed and discussed, and the validity of the optimized MTS with interfacial TME features significantly affecting the load-bearing capacity of the macroscopic properties of the generic slip linear contact is confirmed.
We are very hoped to publish this article in your journal, and I thank you on behalf of our group. We apologize for what we have not done well. We hope we will continue to submit better articles to you.
Thank you in advance for considering this revised submission. We very much look forward to your reply and any questions needed next.
Thank you and best regards.
Sincerely,
Corresponding author/First author
Xigui Wang Professor, PhD Supervisor
School of Engineering Technology, Northeast Forestry University, No. 26, Hexing Road, Xiangfang District, Harbin, 150040, China; School of Mechatronics and Automation, Huaqiao University, No. 668 Jimei Avenue, Jimei District, Fujian Province, Xiamen, 361021, China
Hui Huang
School of Mechatronics and Automation, Huaqiao University, No. 668 Jimei Avenue, Jimei District, Fujian Province, Xiamen, 361021, China;
Jingyu Song
School of Mechatronics and Automation, Huaqiao University, No. 668 Jimei Avenue, Jimei District, Fujian Province, Xiamen, 361021, China;
Yongmei Wang
School of Motorcar Engineering, Heilongjiang Institute of Technology, No. 999, Hongqidajie Road, Daowai District, Harbin, 150036, China; School of Mechatronics and Automation, Huaqiao University, No. 668 Jimei Avenue, Jimei District, Fujian Province, Xiamen, 361021, China
Jiafu Ruan
School of Engineering Technology, Northeast Forestry University, No. 26, Hexing Road, Xiangfang District, Harbin, 150040, China; School of Mechatronics and Automation, Huaqiao University, No. 668 Jimei Avenue, Jimei District, Fujian Province, Xiamen, 361021, China

Round 5
Reviewer 1 Report
All concerns have been accounted for.
Author Response
February 4, 2023
Dear Academic Editor
Dear Editor-in-Chief
The open access journal Lubricants
We have submitted a research article hoping to be published in the journal Lubricants, titled “Numerical Analysis of Friction Reduction and ATSLB Capacity of Lubricated MTS with Textured Micro-Elements.” The paper was coauthored by XiguiWang, Hui Huang, Jingyu Song, Yongmei Wang, and Jiafu Ruan. We have checked the manuscript and revised it according to the comments. Overall, the comments have been fair, encouraging and constructive. We have learned much from it. We submit here the revised manuscript to meet the evaluation conditions and requirements of the reviewers.
Response to Reviewers' Comments:
Academic Editor:
On behalf of all members of our team, I would like to thank the Academic Editors for their sound suggestions and the authors have once again thoroughly reviewed and refined this revised manuscript and have reflected the revised content in this paper. In the long term future, our research team will submit more excellent, high-quality articles to your journal.
Academic Editor Notes
The authors' understanding of the long previous work in this area seems to be very limited. There are many works that have done more in the area. I do not see the novelty of the work. It seems like it needs to do a better job of recognizing previous work and framing it within that work. Here are some if these papers (and there are many, many more).
Greco, Aaron, Oyelayo Ajayi, and Robert Erck. "Micro-Scale surface texture design for improved scuffing resistance in gear applications." In International Design Engineering Technical Conferences and Computers and Information in Engineering Conference, vol. 54853, pp. 579-584. 2011.
Shinkarenko, A., Kligerman, Y. and Etsion, I., 2009. The effect of surface texturing in soft elasto-hydrodynamic lubrication. Tribology International, 42(2), pp.284-292.
Shinkarenko, A., Y. Kligerman, and I. Etsion. "The effect of elastomer surface texturing in soft elasto-hydrodynamic lubrication." Tribology letters 36 (2009): 95-103.
Duvvuru, R. S., Jackson, R. L., & Hong, J. W. (2008). Self-adapting microscale surface grooves for hydrodynamic lubrication. Tribology transactions, 52(1), 1-11.
Jackson, Robert L., and Jiang Lei. "Hydrodynamically lubricated and grooved biomimetic self-Adapting surfaces." Journal of Functional Biomaterials 5, no. 2 (2014): 78-98.
Fesanghary, M., and M. M. Khonsari. "On self-adaptive surface grooves." Tribology transactions 53, no. 6 (2010): 871-880.
Etsion, Izhak. "Improving tribological performance of mechanical components by laser surface texturing." Tribology letters 17 (2004): 733-737.
Etsion, I. (2005). State of the art in laser surface texturing. J. Trib., 127(1), 248-253.
Mao, B., Siddaiah, A., Liao, Y., & Menezes, P. L. (2020). Laser surface texturing and related techniques for enhancing tribological performance of engineering materials: A review. Journal of Manufacturing Processes, 53, 153-173.
Riveiro, A., Maçon, A.L., del Val, J., Comesaña, R. and Pou, J., 2018. Laser surface texturing of polymers for biomedical applications. Frontiers in physics, 6, p.16.
Kovalchenko, A., Ajayi, O., Erdemir, A., Fenske, G., & Etsion, I. (2005). The effect of laser surface texturing on transitions in lubrication regimes during unidirectional sliding contact. Tribology International, 38(3), 219-225.
Braun, Daniel, Christian Greiner, Johannes Schneider, and Peter Gumbsch. "Efficiency of laser surface texturing in the reduction of friction under mixed lubrication." Tribology international 77 (2014): 142-147.
Hsu, Chia-Jui, Andreas Stratmann, Simon Medina, Georg Jacobs, Frank Mücklich, and Carsten Gachot. "Does laser surface texturing really have a negative impact on the fatigue lifetime of mechanical components?." Friction 9 (2021): 1766-1775.
Wang, X., Zhang, H., & Hsu, S. (2007, January). The effects of dimple size and depth on friction reduction under boundary lubrication pressure. In International Joint Tribology Conference (Vol. 48108, pp. 909-911).
Greco, Aaron, Steven Raphaelson, Kornel Ehmann, Q. Jane Wang, and Chih Lin. "Surface texturing of tribological interfaces using the vibromechanical texturing method." Journal of manufacturing science and engineering 131, no. 6 (2009).
Sincerely respond to the Notes raised by Academic Editor:
The authors have made revisions as suggested by Academic Editor and reflected in this manuscript.
As a core member of the team, Professor Wang Xigui has more than 30 years of academic research experience in this field, especially in the research of the correlation mechanism between the lubrication of meshing gear subsets and their anti-thermal elasticity gluing load bearing in heavy-duty high-speed Warship Power Rear Transmission System. A contact interface removal method based on ferrous sulfur oxide bacteria is proposed to investigate the effect of MTS multiscale TME size on the quality of MTS microtexture morphology and lubrication performance, frictional wear behaviour and load bearing capacity by ferrous sulfur oxide bacteria. The experimental process of this study is briefly introduced here to reflect the novelty of this work. We will be submitting the results of the next study to your journal for review and look forward to your suggestions.
The authors have added some relevant literature references to this topic in this revised manuscript, combined with the reasonable comments of the academic editors, and have gained a lot of academic guidance in these references.
Novelty of the research on this topic:
(1) Assuming that multi-scale texturing interfaces are microscopically homogenous and periodic, a formal approach to decoupling macroscopic and microscopic scales is proposed, introducing a micro TEHL dynamic-pressure contact model for interfaces with micro-element homogenization, taking into account non-negligible deformations due to compressive load-bearing and thermal effects at the microscopic scale, and extending the general applicability of classical progressive homogenization methods.
(2) The correlation effect of MTS texturization on the tribological performance of TEHL slip linear contact is investigated through simulations and numerical computational modelling to analyse and predict the load-bearing capacity of MTS with and without interfacial TME features, which is used to assess the friction coefficient of micro-textured MTS.
(3) A pre-designed micro-element texturing patterns at the slip line contact interface under thermoelastic lubrication conditions reveal the correlation between friction and load-bearing capacity of key parameters such as depth and width under mixed thermoelastic hydrodynamic lubrication. The search for critical optimal micro-element shape design parameters and the correlation of geometric scale parameters is necessary to obtain maximum frictional wear reduction benefits and thus improve performance and pursue this concept in gear meshing interfaces for lubrication bearing engineering applications.
Relevant references added in the revised manuscript:
- Duvvuru R. S., Jackson R. L., Hong J. W. Self-adapting microscale surface grooves for hydrodynamic lubrication. Tribology transactions 2008, Vol. 52, Issue 1, p. 1-11. (Corresponds to Reference 2 in this revised manuscript)
- Braun D., Christian G., Johannes S., Peter G. Efficiency of laser surface texturing in the reduction of friction under mixed lubrication. Tribology International 2014, Vol. 17, p. 142-147. (Corresponds to Reference 4 in this revised manuscript)
- Jackson R. L., Jiang L. Hydrodynamically Lubricated and Grooved Biomimetic Self-Adapting Surfaces. Journal of Functional Biomaterials 2014, Vol. 5, Issue 2, p. 78-98. (Corresponds to Reference 6 in this revised manuscript)
- Riveiro A., Maçon A.L., del Val J., Comesaña R., Pou J. Laser surface texturing of polymers for biomedical applications. Frontiers in physics 2018, Vol. 6, p. 16. (Corresponds to Reference 11 in this revised manuscript)
- Mao B., Siddaiah A., Liao Y., Menezes P. L. Laser surface texturing and related techniques for enhancing tribological performance of engineering materials: A review. Journal of Manufacturing Processes 2020, Vol. 53, p. 153-173. (Corresponds to Reference 16 in this revised manuscript)
- Hsu C. J., Andreas S., Simon M., Georg J., Frank M., Carsten G. Does laser surface texturing really have a negative impact on the fatigue lifetime of mechanical components? Friction 2021, Vol. 9, p. 1766-1775. (Corresponds to Reference 41 in this revised manuscript)
Added expressions are illustrated in this revised manuscript with 3 examples as follows.
Example 1: Numerical methods are adopted to model and simulate the lubricated performance by inducing changes in the multi-microscale geometry of the contact interface. The multi-microscale structures perform better in terms of the anti-scuffing load-bearing capacity for the same amount of oil film thickness [2].
Example 2: Numerical approach is based on the deformability of biomaterials to investigate the adaptive surfaces load-bearing properties. [3-5].
Example 3: Practical engineering applications of MTS texturing in gear transmission system to re-duce friction, control wear, improve lubrication and enhance ATSLB capability are attracting increasing interest. It is accepted that the optimization of MTS textures should be tailored to the specific requirements of the application.
Contribution of each individual co-author:
No conflict of interest exits in the submission of this manuscript, and manuscript is approved by all authors for publication. We would like to declare on behalf of our co-authors that the work described was original research that has not been published previously, and not under consideration for publication elsewhere, in whole or in part. All the authors listed have approved the manuscript that is enclosed.
In this subject research, we have proposed the “Numerical Analysis of Friction Reduction and ATSLB Capacity of Lubricated MTS with Textured Micro-Elements” for a theoretical model of thermoelastic lubricated interfacial Textured Micro-Element (TME) load-bearing contact, and the effective friction reduction and Anti-Thermoelastic Scuffing Load-Bearing (ATSLB) characteristics between random rough Meshing Teeth Surfaces (MTS) are investigated, the mechanism linking interfacial thermoelastic lubrication, TME meshing friction reduction and ATSLB is revealed. The MTS is pre-set as a line contact mode, which breaks through the limitation of assuming that the actual meshing area is much smaller than the nominal interface contact domain. The real contact domain area between MTS with multi-scale Micro-Element Textures (MET) is obtained for the numerical calculation of the three-dimensional equivalent TME contact volume, which is the correlation bridge between friction reduction and ATSLB of the thermoelastic lubrication interface.
Our main contribution to the field is to predict the time-varying behaviour of the textured meshing interface friction reduction with TME contact load under thermoelastic lubrication conditions by the proposed theoretical model. Numerical simulations show that the textured interface meshing volume is the key to solving the load-bearing problem of line contact between randomly rough teeth surfaces. The friction coefficients of the MTS are reduced by 13-24%. Interfacial MET parameters are optimized and the correlation between linear velocity, time-varying load and micro-configuration scale and meshing interface oil film thickness, oil film pressure distribution morphological trends and ATSLB capacity is elaborated. The lubricated load-bearing and friction reduction behaviour between the textured MTS is quantified by the thermoelastic voids of the TME interface and the actual meshing volume ratio, which provides a new perspective for further insight into the lubrication and friction reduction behaviour between the MTS with multi-scale MET-ATSLB coupling mechanism.
The novelty and significance of this manuscript are as follows:
The work was studied mainly “Numerical Analysis of Friction Reduction and ATSLB Capacity of Lubricated MTS with Textured Micro-Elements”. This topic involves the homogenized micro hydrodynamics of the MTS with interfacial TME characteristics in the TEHL steady state is approximated accurately, the microgeometry of the multiscale structure with contact interface TME features is simulated, the existence of an optimal geometry that is particularly effective for the load-bearing capacity of the MTS is proposed and discussed, and the validity of the optimized MTS with interfacial TME features significantly affecting the load-bearing capacity of the macroscopic properties of the generic slip linear contact is confirmed.
We are very hoped to publish this article in your journal, and I thank you on behalf of our group. We apologize for what we have not done well. We hope we will continue to submit better articles to you.
Thank you in advance for considering this revised submission. We very much look forward to your reply and any questions needed next.
Thank you and best regards.
Sincerely,
Corresponding author/First author
Xigui Wang Professor, PhD Supervisor
School of Engineering Technology, Northeast Forestry University, No. 26, Hexing Road, Xiangfang District, Harbin, 150040, China; School of Mechatronics and Automation, Huaqiao University, No. 668 Jimei Avenue, Jimei District, Fujian Province, Xiamen, 361021, China
Hui Huang
School of Mechatronics and Automation, Huaqiao University, No. 668 Jimei Avenue, Jimei District, Fujian Province, Xiamen, 361021, China;
Jingyu Song
School of Mechatronics and Automation, Huaqiao University, No. 668 Jimei Avenue, Jimei District, Fujian Province, Xiamen, 361021, China;
Yongmei Wang
School of Motorcar Engineering, Heilongjiang Institute of Technology, No. 999, Hongqidajie Road, Daowai District, Harbin, 150036, China; School of Mechatronics and Automation, Huaqiao University, No. 668 Jimei Avenue, Jimei District, Fujian Province, Xiamen, 361021, China
Jiafu Ruan
School of Engineering Technology, Northeast Forestry University, No. 26, Hexing Road, Xiangfang District, Harbin, 150040, China; School of Mechatronics and Automation, Huaqiao University, No. 668 Jimei Avenue, Jimei District, Fujian Province, Xiamen, 361021, China
